# PRF: Parallel Resonate and Fire Neuron for Long Sequence Learning in Spiking Neural Networks

## Abstract

Recently, there is growing demand for effective and efficient long sequence modeling, with State Space Models (SSMs) proving to be effective for long sequence tasks. To further reduce energy consumption, SSMs can be adapted to Spiking Neural Networks (SNNs) using spiking functions. However, current spiking-formalized SSMs approaches still rely on float-point matrix-vector multiplication during inference, undermining SNNs' energy advantage. In this work, we address the efficiency and performance challenges of long sequence learning in SNNs simultaneously. First, we propose a decoupled reset method for parallel spiking neuron training, reducing the typical Leaky Integrate-and-Fire (LIF) model's training time from $O(L^2)$ to $O(L \log L)$, effectively speeding up the training by $6.57\times$ to $16.50\times$ on sequence lengths $1,024$ to $32,768$. To our best knowledge, this is the first time that parallel computation with a reset mechanism is implemented achieving equivalence to its sequential counterpart. Secondly, to capture long-range dependencies, we propose a Parallel Resonate and Fire (PRF) neuron, which leverages an oscillating membrane potential driven by a resonate mechanism from a differentiable reset function in the complex domain. The PRF enables efficient long sequence learning while maintaining parallel training. Finally, we demonstrate that the proposed spike-driven architecture using PRF achieves performance comparable to Structured SSMs (S4), with two orders of magnitude reduction in energy consumption, outperforming Transformer on Long Range Arena tasks. [1]

## 1 Introduction

**Long Sequence Modeling.** Long sequence modeling is a fundamental problem in machine learning. This significant advancement has yielded wide-ranging impact across various fields (such as Transformer (Vaswani et al., 2017) and Mamba (Gu & Dao, 2023)), including reinforcement learning (e.g., robotics and autonomous driving) (Chen et al., 2021), autoregressive task (e.g., large language models) (Zhao et al., 2023), generative tasks (e.g., diffusion model) (Peebles & Xie, 2023) and etc. The Transformer architecture (Vaswani et al., 2017), which combines token mixing with self-attention and channel mixing with dense matrices, has been successfully applied in these fields, since sequence data (with length $L$) is well modeled by the self-attention mechanism.

State space models (SSMs) (Gu et al., 2022a) effectively address the limitations of self-attention machanism in processing long sequences by reducing computational complexity from $O(L^2)$ to $O(L)$ during inference (Feng et al., 2024). The recurrent form of SSMs allows themselves to scale to longer sequence lengths more efficiently. Furthermore, the (Structured) SSMs utilize orthogonal polynomial bases to initialize recurrent weights (Gu et al., 2020) for token mixing, and to compress input history (Gu et al., 2021) into hidden state that enables long sequence learning yielding superior performance than Transformer models on long sequence tasks (Gu et al., 2022a). As a result, variants of SSMs (e.g. Mamba (Gu & Dao, 2023)) are successfully applied across various fields. However, SSMs still requires large number of float-point matrix-vector multiplications for both token mixing and channel mixing to effectively extract information. These dense matrix multiplication computations scale quadratically with model size ($D$), consuming significant amount of energy during inference.

---

[1]The GitHub repository will be released after paper accepted.

**Spiking Neural Networks.** SNNs benefit from the ability to convert the Float-Point (FP) Multiply Accumulate (MAC) computations into sparse Accumulate (AC) computations through their spike-driven mechanism (Hu et al., 2021; Yao et al., 2024). Prior arts incorporated the spiking function at the input of token and channel mixing in SSMs to reduce energy consumption in channel mixing (Stan & Rhodes, 2023; Bal & Sengupta, 2024; Shen et al., 2024). However, the token mixing operation within the SSMs blocks has yet been optimized, resulting in extensive FP matrix-vector multiplications and thus compromising the overall energy efficiency (Stan & Rhodes, 2023). To avoid FP matrix-vector multiplication computations in both channel mixing and token mixing, it is also crucial to reduce the complexity of token mixing. Therefore, we must revisit and design novel the spiking neurons in SNNs to enabl effective long sequence learning capabilities, while maintaining computation efficiency.

**The Challenges for SNNs.** There are two challenges in implementing long sequence learning in SNNs. (i) First, the Backpropagation Through Time (BPTT) is the commonly used training method for SNNs (Wu et al., 2018). However, BPTT leads to a quadratic growth in training time with respect to sequence length, scaling as $O(L^2)$ (Kag & Saligrama, 2021). While there are some existing SNN efficient training methods that can improve training efficiency (Bellec et al., 2020; Xiao et al., 2022; Yin et al., 2023), they are still outperformed by BPTT for longer sequences (Meng et al., 2023). (ii) Secondly, commonly used spiking neuron models struggle to capture long-range dependencies. Specifically, the widely used LIF neuron has difficulty in capturing and distinguishing dependencies in membrane potential over long intervals, which limits its performance on long-range tasks.Previous work attempted to address the training efficiency and performance improvement separately by improving neuron models (Fang et al., 2024; Spieler et al., 2024). (Fang et al., 2024; Spieler et al., 2024). However, efforts made to tackle these challenges on long sequence tasks at the same time remains unveiled. In this work, we aim for addressing these two challenges simultaneously. **Our contributions are summarized as follows**:

- To accelerate the training process, we propose a novel decoupled reset method to implement parallel training that is equivalent to sequential training. This approach accelerates the back propagation by three orders of magnitude and can be applied to any types of spiking neurons.
- To effectively extract long range dependencies, we propose the Parallel Resonate and Fire (PRF) neuron with oscillating membrane potential in complex domain leveraging an adaptive and differentiable reset mechanism.
- To minimize inference energy, we further incorporate PRF into the design of Spike-Driven Temporal and Channel Mixer (SD-TCM) module. This module achieves performance comparable to S4 in long range arena tasks while reducing the inference energy by two orders of magnitude.

## 2 RELATED WORK

**State Space Models** A general discrete form of SSMs is given by the equation: $\boldsymbol{u}_t = \boldsymbol{A}\boldsymbol{u}_{t-1} + \boldsymbol{B}\boldsymbol{x}_t$, $\boldsymbol{y}_t = \boldsymbol{C}\boldsymbol{u}_t$, where $\boldsymbol{A} \in \mathbb{R}^{H \times H}$, $\boldsymbol{B} \in \mathbb{R}^{D \times H}$, $\boldsymbol{C} \in \mathbb{R}^{H \times D}$ matrices is shape with the model size $D$ and hidden size $H$. The success of the Structured SSMs (S4) arises from the fact that the coefficients of the orthogonal bases are solved to fit an arbitrary sequence curve (Gu et al., 2020). By leveraging orthogonal polynomial bases for initializing structured matrices $\boldsymbol{A}$ and $\boldsymbol{B}$, S4 effectively compresses input history and outperforms transformers in long-range sequence tasks (Gu et al., 2021). Subsequent variants of SSMs have also achieved great success on long sequence task (Gu et al., 2022a; Goel et al., 2022b; Gu et al., 2023; 2022b; Orvieto et al., 2023). However, these SSMs-based model still exist power consumption issues that scale quadratically with model size $D$. This is largely due to the use of the dense matrices for channel mixing after $\boldsymbol{y}_t$, such as in the GLU block (Dauphin et al., 2017), which requires $2D^2$ computation in matrices, leading to a significant number of FP-MAC operations.

**Spikinglized SSMs** The spike mechanism can alleviate the energy problem in dense matrices by converting the FP-MAC as sparse FP-AC computation. Some spiking-formalized (spikinglized) approaches integrate the spike function into SSMs after token mixing to reduce the FP-MAC computation of channel mixing. For instance, Oliver et al.(Stan & Rhodes, 2023) make intersection of SNNs with S4D model (Gu et al., 2022b) for long-range sequence modelling, by adding Heaviside function to token mixing output, $\boldsymbol{y}_t$, at each SSMs layer. Similarly, Abhronil et al.(Bal & Sengupta,

2024) combine stochastic spiking function at the output of $\boldsymbol{y}_t$. These spikinglized method can harvest the long sequence learning capability of SSMs models, but also retain nonlinear activation computation and FP matrix-vector multiplications, as they retain the $\boldsymbol{A}$, $\boldsymbol{B}$ and $\boldsymbol{C}$ matrices during recurrent inference. However, this retention limits the energy efficiency advantages of SNNs and poses challenges for deploying the model on neuromorphic chips. Therefore, we aim to further optimize the inference process by reducing matrices $\boldsymbol{A}$ and $\boldsymbol{B}$ to vectors and eliminating $\boldsymbol{C}$. This requires rethinking the role of spiking neurons in long sequence learning.

## 3 PROBLEM FORMULATION

In this section, we first introduce the commonly used Leaky Integrate-and-Fire (LIF) neuron, then we describe the two main challenge for long range learning ability with spiking neuron.

### 3.1 THE LEAKY INTEGRATE-AND-FIRE (LIF) NEURON

The LIF model is a widely used spiking neuron model. It simulates neurons by integrating input signals and firing spikes when the membrane potential exceeds a threshold. The dynamic of membrane potential $u(t)$ is followed by:

$$\frac{du(t)}{dt} = -\frac{1}{\tau}\left(u(t) - u_{\text{reset}}\right) + \frac{R}{\tau}c(t), \tag{1}$$

where the $c(t)$ is the input current. The constants $\tau$, $u_{\text{reset}}$, and $R$ denote the membrane time constant, reset potential, and resistance. We use $R = \tau$ as in previous work. When $u(t)$ reaches the threshold $V_{\text{th}}$, the neuron fires a spike, and then $u(t)$ resets to $u_{\text{reset}}$. The discrete LIF model is expressed as:

$$u_t = \beta \cdot \left(u_{t-1} - V_{\text{th}}s_{t-1}\right) + c_t, \tag{2}$$

$$s_t = \mathcal{H}(u_t - V_{\text{th}}) = \begin{cases} 1, & \text{if } u_t \geq V_{\text{th}} \\ 0, & \text{otherwise} \end{cases}, \tag{3}$$

where $\beta \triangleq 1 - \frac{1}{\tau} \in (0, 1)$, and the discrete timesteps $t = 1, 2, \ldots, T$, the initial situation $s_0 = u_0 = 0$. After firing, the membrane potential resets according to previous spike $s_{t-1}$. This spiking neuron face two primary challenges for long sequence tasks: (i) The coupled reset prevents parallel training along timesteps, causing training time significantly for long sequences. (ii) The commonly used LIF neuron model struggles to capture long-range dependencies, hindering the performance on long sequence tasks. The overview as shown in Figure 1.

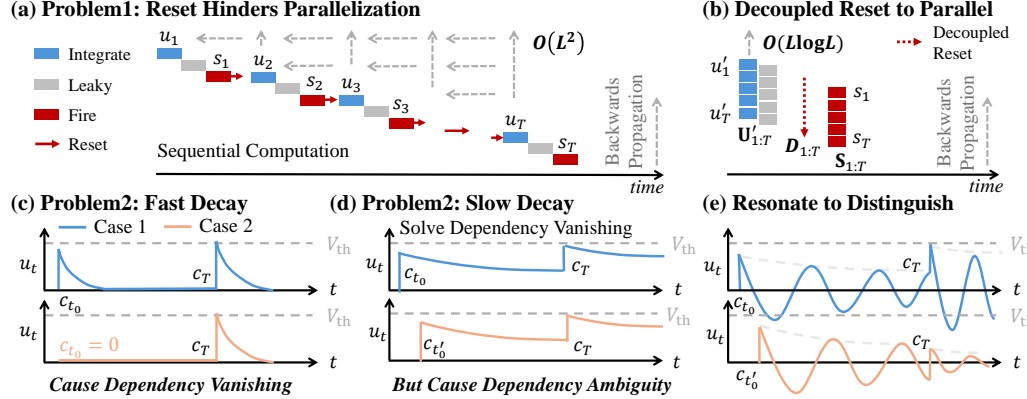

Figure 1: (a) The reset mechanism prevents parallel computation of timesteps as the state update relies on previous spike output, causing $O(L^2)$ timing cost. (b) The proposed decoupled reset mechanism enables parallel computation. (c) Fast decay causes long-range dependencies to vanish in the membrane potential. (d) Slow decay could generate dependency over long sequence, but causes dependency ambiguity. (e) The resonate mechanism with membrane potential helps distinguish relevant inputs.

### 3.2 PROBLEM 1: COUPLED RESET PREVENTS PARALLELISM ALONG TIMESTEPS

The first problem is that the training cost of LIF-based sequential computation increases exponentially as the number of timesteps extends, with a complexity of $O(L^2)$. This occurs because BPTT method requires the computational graph to expand along the time dimension (Kag & Saligrama, 2021). While the sequential computation of linear combination can be parallelized to reduce the time cost complexity from $O(L^2)$ to $O(LlogL)$ during training, as done in SSMs. However, the $u_t$ dependency on previous spikes output $s_{t-1}$ with nonlinear Heaviside function. By recursively expanding $u_t$ and simplifying in Equation 4:

$$u_t = c_t + \beta c_{t-1} + ... + \beta^{t-1}c_1 - \beta V_{\text{th}}(s_{t-1} + s_{t-2} + ... + s_1). \tag{4}$$

The reset mechanism causes $u_t$ to depend on all previous spike outputs. This coupling hinders the neuron's ability to perform parallel computations, further limiting effective parallel computation for both $u_t$ and $s_t$, as illustrated in Figure 1 (a). This forces the spiking neuron to compute sequentially. Consequently, as the number of timesteps increases during training, the time cost grows quadratically. To address this issue, we propose the decoupled reset method, which facilitates parallel computation of spiking neurons with the reset mechanism (Illustrated in Figure 1 (b) and discussed in detail in Method 4.1).

### 3.3 PROBLEM 2: COMMONLY USED LIF STRUGGLE WITH LONG-RANGE DEPENDENCIES

The second problem is that the commonly used LIF model struggles to capture and distinguish the dependencies in the membrane potential over long interval. This limitation arises due to the dilemma of decay factor $\beta$ in the membrane potential dynamics. When $\beta$ is small, the membrane potential decreases rapidly. This **Fast Decay** cause the neuron to quickly forget past inputs (Figure 1 (c)). For example, in Case 1, if there are two inputs, $c_{t_0}$ and $c_T$, separated by $T$ time steps, result in $u_T$ becoming independent of $c_{t_0}$. Similarly, in Case 2, with zero input $c_{t_0}$, the result is identical with Case 1. This failure to retain long-term information leads to the vanishing of dependencies. Conversely, when $\beta$ is large, the membrane potential retains its value over an extended period $T$ (Figure 1 (d)). This **Slow Decay** can solve the dependency vanishing problem, but causing ambiguity between closely spaced inputs. For instance, input $c_{t_0} = c_{t'_0}$ with $t'_0 = t_0 + \delta t$ result in similar membrane potentials, there is small difference between the membrane potential $u_T = \beta^{(T)} \cdot c_{t_0} + c_T$ in Case 1 and the $u_T = \beta^{(T-\delta t)} \cdot c_{t'_0} + c_T$ in Case 2, as shown in Figure 1 (d). To address this challenge, we propose enhancing the neuron's dynamics by incorporating resonate mechanism. This allows neurons to remain sensitive to input, even after a long interval of $T$. (Illustrated in Figure 1 (e) and discussed in detail in Method 4.2).

## 4 METHOD

In this section, we present our approach to address two primary challenges in SNNs: parallel training and long-range dependency learning. For parallel training, we decouple the reset mechanism from the integrate computation to implement parallel training while maintaining equivalence with sequential computation. For long-range dependency learning, we propose the Parallel Resonate and Fire (PRF) Neuron by incorporating the reset process as an imaginary part into the time constant, enabling the neuron to achieve long-range learning ability.

### 4.1 DECOUPLED RESET FOR PARALLEL COMPUTATION

To enable parallel computation, we need to decouple the causal relationship with previous spikes, by separating the linear combination part from the nonlinear causal dependency part. To achieve this, we first substitute the Equation 4 into the Equation 3 by expanding the $u_t$ in the Heaviside function. Then we merge the reset part into the threshold $V_{\text{th}}$. As such, Equation 3 is rewritten as Equation 5:

$$s_t = \mathcal{H}(\underbrace{c_t + \beta c_{t-1} + ... + \beta^{t-1}c_1}_{\text{Leaky and Integrate} \triangleq u'_t} - \underbrace{V_{\text{th}} \cdot (\beta \cdot (s_{t-1} + s_{t-2} + ... + s_1) + 1)}_{\text{decoupled reset} \triangleq d_t})), \tag{5}$$

where the linear combination of leaky and integrate part is defined as $u_t'$, the second part is defined as the decoupled reset $d_t$. Now the spike output is:

$$s_t = \mathcal{H}\left(u_t' - d_t\right) = \begin{cases} 1, & \text{if } u_t' \geq d_t \\ 0, & \text{otherwise} \end{cases}, \tag{6}$$

where this spike output $s_t$ depends on $u_t'$ and $d_t$. Note that the first part is linear combination of $u_t' = \beta u_{t-1}' + c_t$, we define the sequence of $\mathbf{U}_T'$ as the ordered set $\{u_1', u_2', ..., u_T'\}$ over all timesteps (detailed notation is described in Appendix A). This linear combination of $u_t'$ can be computed using convolution and further accelerated by converting the convolution into multiplication after applying the Fast Fourier Transform: $\mathbf{U}_T' = \mathbf{C}_T * \mathbf{K}_T = \mathcal{F}^{-1}\left(\mathcal{F}(\mathbf{C}_T) \odot \mathcal{F}(\mathbf{K}_T)\right)$, with the $O(L log L)$ computation complexity, avoiding the $O(L^2)$ training time in BPTT.

After efficient parallel computation of $u_t'$, the $d_t$ still remains the dependency relationship with previous spikes $s_t$. To further decouple this dependency from the spike output $s_t$, the key idea is **converting the recursive form to the iterative form**. We first define the dependency part as $A_t$, as shown in Equation 7. Now, we only need to convert the $A_t$ from its recursive form into an iterative form. By separating the last spike from all previous spikes, we obtain Equation 8. The left part of this equation corresponds to Equation 6, while the right part refers to the recursive dependency itself, leading to Equation 9 and Equation 10:

$$A_t \triangleq s_{t-1} + s_{t-2} + ... + s_1 \tag{7}$$
$$= s_{t-1} + (s_{t-2} + ... + s_1) \tag{8}$$
$$= \mathcal{H}\left(u_{t-1}' - d_{t-1}\right) + A_{t-1} \tag{9}$$
$$= \begin{cases} 1 + A_{t-1}, & \text{if } u_{t-1}' \geq d_{t-1} \\ A_{t-1}, & \text{if } u_{t-1}' < d_{t-1} \end{cases}, \tag{10}$$

where the sequence of $d_t$ is calculated according to the sequence of $u_t'$, by dynamically updating $A_t$:

$$d_t = V_{\text{th}} \cdot (\beta A_t + 1), \quad A_0 = u_0' = 0. \tag{11}$$

At this point, all recursive forms have been converted into dynamic equations. The formation of $d_t$ is completely independent of $s_t$. The decoupled reset function $\mathbf{D}_T = f_D(\mathbf{U}_T')$ is deduced as shown in Equation 10 and 11, where $d_t$ can be dynamically scanned from all $u_t'$ with $O(L)$ complexity.

In summary, we convert the all calculation of $s_t$ with $O(L^2)$ complexity into a combination of $u_t'$ with $O(L \log L)$ complexity and subsequently $d_t$ with $O(L)$ complexity, achieving a training speed-up of approximately $L/(\log L + 1)$. To summarize, this approach facilitates parallel computation:

$$\mathbf{U}_T' = \mathbf{C}_T * \mathbf{K}_T \tag{12a}$$
$$= \mathcal{F}^{-1}\left(\mathcal{F}(\mathbf{C}_T) \odot \mathcal{F}(\mathbf{K}_T)\right) \tag{12b}$$
$$\mathbf{D}_T = f_D(\mathbf{U}_T') \tag{12c}$$
$$\mathbf{S}_T = \mathcal{H}(\mathbf{U}_T' - \mathbf{D}_T) \tag{12d}$$

$$\mathbf{U}_T' = (u_1', u_2', \ldots, u_T') \tag{13a}$$
$$\mathbf{C}_T = (c_1, c_2, \ldots, c_T) \tag{13b}$$
$$\mathbf{K}_T = (\beta^0, \beta^1, \ldots, \beta^{T-1}) \tag{13c}$$
$$\mathbf{D}_T = (d_1, d_2, \ldots, d_T) \tag{13d}$$

Where the outputs $\mathbf{S}_T = (s_1, s_2, \ldots, s_T)$ from Equation 12d is equivalent with the sequential generated from Equation 3. The kernel vector $\mathbf{K}_T$ with $\beta = 1 - \frac{1}{\tau}$. By combining the above equations, we obtain the parallel computation process, as shown in Algorithm.1 and 2 in Appendix.B. Although parallelized LIF solves the training problem for long sequences (Experiment 5.1), it still does not perform well on long sequences (Experiment 5.2), due to the common issue of long-range dependencies in LIF models (Problem 3.3).

## 4.2 RESONATE FOR LONG-RANGE LEARNING ABILITY

To address the challenge of capturing long-range dependencies in spiking neural networks, we introduce the *Parallel Resonate-and-Fire* (PRF) neuron. This neuron model extends the standard LIF neuron by incorporating a resonance mechanism into its dynamics, enabling it to retain information over longer periods while maintaining computational efficiency.

Firstly, recall the commonly used LIF model from Equation 1, the dynamics are given by: $\frac{du(t)}{dt} = \gamma u(t) - \gamma u_{\text{reset}} + c(t)$, where $\gamma \triangleq -1/\tau$. To introduce more dynamic behavior, we use $r(t)$ and $\theta$

to replace $u_{\text{reset}}$ and $\gamma$ in the reset part, in respectively. Then the dynamic are rewritren as:

$$\frac{du(t)}{dt} = \gamma u(t) - \theta r(t) + c(t). \tag{14}$$

To introduce membrane potential oscillations, we define a complex decay constant $\tilde{\gamma} = \gamma + i\theta$, where $i = \sqrt{-1}$. In the vanilla LIF model, the reset $r(t)$ is a non-continuous *conditional function* with $\theta = \gamma$ in Equation 14, and the reset result is instantaneous process:

$$\frac{dr(t)}{dt} = \begin{cases} V_{\text{th}}\delta(t), & \text{if } u(t) \geq V_{\text{th}} \\ 0, & \text{if } u(t) < V_{\text{th}} \end{cases}. \tag{15}$$

While this reset has proven effective, it poses challenges for efficient computation. We introduce a reset function that is both effective and computationally simple for both forward and backward propagation. We define the reset function to satisfy the condition: $\tilde{u}(t) \triangleq u(t) + ir(t)$. Substituting this definition into Equation 14, we obtain the PRF model:

$$\frac{d\tilde{u}(t)}{dt} = \tilde{\gamma}\tilde{u}(t) + c(t), \tag{16}$$

where the real part $\Re\{\tilde{u}(t)\}$ corresponds to the membrane potential dynamics. This formulation incorporates the reset mechanism into the time constant via the imaginary component $\theta$, introducing resonance into the neuron's behavior. When $\theta = 0$, the model reduces to the standard LIF neuron without reset. If we extend the Equation 16, we can be expressed in matrix form:

$$\begin{pmatrix} \dot{u}(t) \\ \dot{r}(t) \end{pmatrix} = \begin{pmatrix} \gamma & -\theta \\ \theta & \gamma \end{pmatrix} \begin{pmatrix} u(t) \\ r(t) \end{pmatrix} + \begin{pmatrix} c(t) \\ 0 \end{pmatrix}, \tag{17}$$

which give us the insight that this reset process is continuous function:

$$\frac{dr(t)}{dt} = \gamma u(t) + \theta r(t), \tag{18}$$

where this formulation treats the reset as a continuous function controlled by $\theta$, allowing for a more gradual and reset process.

We then discretize the model (Equation 16) (details are provided in Appendix C), yielding the sequential and parallel formulations of the PRF neuron, as shown in Equation 19 and 20 in respectively:

$$\tilde{u}_t = \exp(\Delta\tilde{\gamma})\tilde{u}_{t-1} + \Delta c_t \quad \text{(19a)} \qquad\qquad \tilde{\mathbf{U}}_T = \mathcal{F}^{-1}\left(\mathcal{F}\left(\mathbf{C}_T\right) \odot \mathcal{F}(\tilde{\mathbf{K}}_T)\right) \quad \text{(20a)}$$

$$s_t = \mathcal{H}\left(\Re\{\tilde{u}_t\} - V_{\text{th}}\right) \quad \text{(19b)} \qquad\qquad \mathbf{S}_T = \mathcal{H}\left(\Re\{\tilde{\mathbf{U}}_T\} - V_{\text{th}}\right) \quad \text{(20b)}$$

$$\tilde{\gamma} = \gamma + i\theta \quad \text{(19c)}$$

Now the membrane potential $\tilde{u}_t \in \mathbb{C}$, where $\Delta$ is the time step size. The details of parallel computation is described in Algorithm.3. Where $\mathbf{C}_T = (c_1, c_2, \ldots, c_T)$ is the input current sequence, and the kernel $\tilde{\mathbf{K}}_T$ is calculated by:

$$\tilde{\mathbf{K}}_T = \left(\Delta A^{(0)}, \Delta A^{(1)}, \ldots, \Delta A^{(T-1)}\right), \tag{21}$$

with $A = \exp\left(\Delta\tilde{\gamma}\right)$. By incorporating the imaginary component $\theta$, the neuron exhibits oscillatory behavior, allowing it to resonate at specific input frequencies (see Appendix D for details). This oscillation phenomenon can be found in biological neurons (Izhikevich, 2001). Our model differs from variants of resonant models, like BHRF (Higuchi et al., 2024), which combine adaptive thresholds and refractory mechanisms. Instead, we incorporate the reset directly into the time constant through the imaginary component while enabling efficient parallel computation for training. Furthermore, the PRF neuron can also be deployed on neuromorphic chips for inference after parallel training, requiring only two additional multiplications and one more addition than LIF (see details in Appendix E).

### 4.3 THEORY ANALYSIS

*(1) Sequential Perspective on Dynamic.* Decoupling the reset can be viewed as transforming the reduction in membrane potential caused by increased threshold value. This means that the soft reset mechanism could be equivalent to the adaptive threshold mechanism (Bellec et al., 2020) as deduced in the Theorem 1. Furthermore, the LIF model without a reset can be considered a specific instance of a PRF, as demonstrated in Theorem 2. Moreover, the $u_t$ in PRF will converge as shown in Theorem 3. This indicates that the membrane potential of PRF is stable and bounded.

**Theorem 1.** *Let $V_{\text{th}} = 1$ and $\rho = 1$ in Adaptive-LIF model, then the LIF neuron with soft reset model is equivalent to the Adaptive-LIF without reset mechanism. (The proof See Appendix.F.1)*

**Theorem 2.** *Let $\Delta = 1$ and $\theta = 0$, then the PRF model could degenerate as Valina LIF model without reset mechanism. (The proof see Appendix.F.2)*

**Theorem 3.** *If the inputs $c_t \sim \mathcal{N}(0, \sigma^2)$ follow a normal distribution, then the membrane potential of PRF will converge as a distribution $u_t \sim \mathcal{N}(0, \frac{\tau\Delta}{2}\sigma^2)$, as $t \to +\infty$. (The proof see Appendix.F.3)*

*(2) Parallel Perspective on Gradients.* The issue of membrane potential dependence described in Problem 3.3 is fundamentally a problem of gradients, which are correlated with the kernel and previous spikes. Assuming the input current is $\mathbf{C}_T^l = \boldsymbol{W}^l \mathbf{S}_T^{l-1}$ across all $T$, the gradients are proportional to:

$$\nabla_{\boldsymbol{W}^l}\mathcal{L} \propto \underbrace{\frac{\partial \mathcal{L}}{\partial u_T^l} \sum_{t=1}^{T} \frac{\partial u_T^l}{\partial u_t^l} \frac{\partial u_t^l}{\partial \boldsymbol{W}^l}}_{\text{Sequential Perspective}} \propto \underbrace{\sum_{t=1}^{T} \beta^{(t)} s_t^{l-1} = \left\langle \mathbf{K}_T, \mathbf{S}_{T:1}^{l-1} \right\rangle}_{\text{Parallel Perspective}}, \tag{22}$$

here $\langle \cdot, \cdot \rangle$ denotes the inner product, $\mathbf{S}_T^{l-1} = (s_1^{l-1}, s_2^{l-1}, \ldots, s_T^{l-1})$ is the sequence of spike outputs from the previous layer $l - 1$, and $\mathbf{K}_T$ represents the parallelism kernel. The subscript $(T : 1)$ indicates the sequence reversal. A small $\beta$ causes gradient vanishing due to a narrow receptive field, making it likely for sparse spikes to fall outside this field and yield an inner product close to zero. Conversely, a large $\beta$ results in a long-range, slow-decaying kernel, leading to overly consistent gradient values across spike positions. However, an oscillating kernel with a large $\beta$ can adjust the gradient at different spike positions, smoothing the gradients and improving the neuron's representational capability (see Appendix G for details). Experiment 5.2 verifies this insight.

### 4.4 Architecture

The *Spike-Driven Token and Channel Mixer* (SD-TCM) Module (Figure 2, with further details in Appendix H) is inspired by token and channel mixing in Transformer and S4 architectures. For token mixing, we use a PRN Neuron followed by a Linear layer, while for channel mixing, the Neuron is replaced by a Spatial Neuron. The Spatial Neuron, a variant of the LIF Neuron, focuses on instantaneous information at each timestep by setting the time constant $\tau$ close to 1: $\lim_{\tau \to 1^+} u_t = \left(1 - \frac{1}{\tau}\right) u_{t-1} + c_t \approx c_t$. This simplifies the output to $s_t = \mathcal{H}(c_t - V_{\text{th}})$, resembling motor neurons in biology with rapid decay.

During training, a trainable amplitude $\alpha$ acts as a gate, and during inference, $\alpha$ can be merged into the Linear layer: $(\alpha \boldsymbol{s}_t) \times \boldsymbol{W} \equiv \boldsymbol{s}_t \times (\alpha \boldsymbol{W})$. Membrane shortcut residual connections are used to maintain event-driven, spike-based communication. As shown in Table 1, the PRF requires only $O(5D)$ computational complexity, while SSMs and Spikinglized SSMs require $O(H^2 + 2DH)$. This makes the architecture rely only on FP-AC and element-wise multiplication, reducing energy consumption and simplifying deployment on neuromorphic chips. As a result, it achieves lower computational complexity and energy consumption (see analysis in Appendix I).

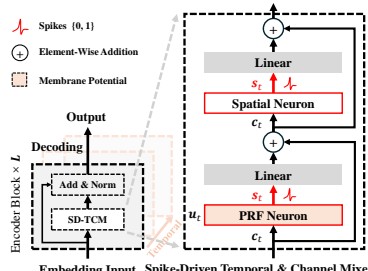

Figure 2: Diagram of the SD-TCM.

Table 1: The comparison for inference complexity. Full table in Appendix.I ($H$: hidden dimension, $D$: model dimension, $\mathcal{H}$ denotes Heaviside function, $fr$: firing rate $\in (0, 1)$).

| Token Mixing | Dynamic Equation | Infer. Complexity |
|---|---|---|
| SSMs | $\boldsymbol{u}_t = \boldsymbol{A}^{H \times H} \boldsymbol{u}_{t-1} + \boldsymbol{B}^{D \times H} \boldsymbol{x}_t$ 
 $\boldsymbol{y}_t = \boldsymbol{C}^{H \times D} \boldsymbol{u}_t$ | $O(H^2 + 2DH)$ |
| Spikinglized SSMs | $\boldsymbol{u}_t = \boldsymbol{A}^{H \times H} \boldsymbol{u}_{t-1} + \boldsymbol{B}^{D \times H} \boldsymbol{x}_t$ 
 $\boldsymbol{y}_t = \mathcal{H}(\boldsymbol{C}^{H \times D} \boldsymbol{u}_t - V_{\text{th}})$ | $O(H^2 + 2DH)$ |
| PRF + Linear | $\boldsymbol{u}_t = \boldsymbol{a}^D \odot \boldsymbol{u}_{t-1} + \boldsymbol{b}^D \odot \boldsymbol{x}_t$ 
 $\boldsymbol{s}_t = \mathcal{H}(\Re\{\boldsymbol{u}_t\} - V_{\text{th}})$ 
 $\boldsymbol{y}_t = \textbf{Linear}(s_t)$ | $O(5D + fr \cdot D^2)$ |

## 5 EXPERIMENTS

To evaluate the efficiency and effectiveness of the parallel method, as well as the performance improvement on long sequence tasks, we first demonstrate that parallel significantly accelerates training while maintaining equivalence with sequential computation. Next, we explore the PRF neuron's ability to handle long-range dependencies, showing that kernel oscillations improve both performance and gradient stability. Finally, the SD-TCM module achieves performance comparable to S4 while reducing energy consumption by over 98.57% on Long Range Arena tasks. Detailed experimental setups and training hyperparameters are provided in Appendix J.

### 5.1 THE PARALLEL PROCESS

First, we compare the training runtime across different timesteps in Figure 3 *(left)* using three repeated experiments. Beyond 4 timesteps, parallel training consistently outperforms sequential training. For example, at 1,024 timesteps, sequential training takes 4.6 seconds per iteration, while parallel training takes only 0.7 seconds, achieving a speedup of $6.57\times$.

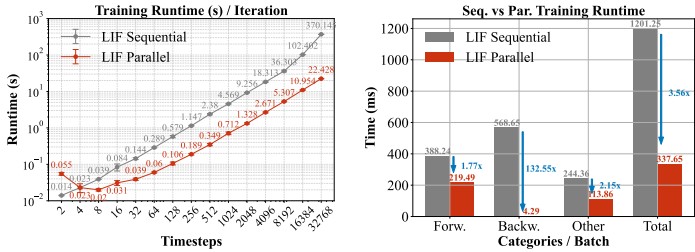

Figure 3: *(left)* Comparison of training runtime for LIF and Parallelized LIF models. *(right)* Comparison of sequential and parallel training runtime across different categories per batch.

This acceleration becomes more significant as the number of timesteps increases, with a speedup of $9.35\times$ at 16,384 timesteps and $16.50\times$ at 32,768 timesteps. The equivalence between sequential and parallel computation is maintained during inference and training, with only a minor accuracy difference during training (details in Appendix K), which we tentatively attribute to numerical error.

The speedup is primarily due to parallel training avoiding the recursive unfolding of the computational graph. The forward and backward passes are accelerated by $1.77\times$ and $132.55\times$, respectively, as shown in Figure 3 *(right)*. The significant speedup in the backward pass occurs because sequential training requires unfolding the graph at each timestep, which is time-consuming, whereas parallel training computes the graph only once. Further details of the timing for both training and inference can be found in Appendix L.

### 5.2 THE LONG RANGE LEARNING ABILITY

Although parallelized LIF can speed up training, it still struggles with performance on simple long-sequence tasks, such as sequential MNIST, as shown in Figure 4 *(left)*. However, after introducing oscillations in the kernel, the PRF neuron successfully solves the sequential-MNIST problem, achieving both training efficiency and effectiveness.

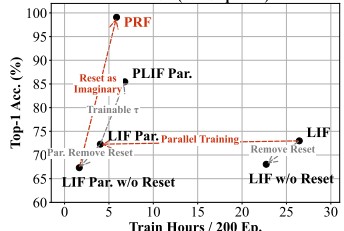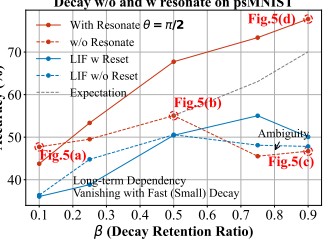

Figure 4: *(left)* Ablation Study on sMNIST datasets (Par. means parallel training). *(right)* Accuracy across different decay on psMNIST.

To better understand how oscillating membrane potential improves performance, we compare the performance of various $\beta$ values, with and without oscillations, on the more challenging permuted-sMNIST dataset. The results after training for 50 epochs are summarized in Figure 4 *(right)*. (We fit $\Delta = 1$ and set $\theta = \frac{\pi}{2}$ while varying the $\beta$ hyperparameter without training.) Without the oscillating term, as $\beta$ increases, accuracy initially improves as expected but then decreases beyond a certain point. In contrast, the introduction of oscillations helps counteract this decline and further enhances performance, indicating that oscillations can improve performance for long sequences.

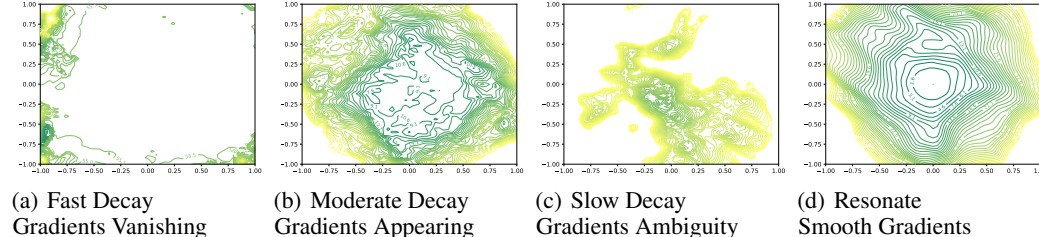

(a) Fast Decay
Gradients Vanishing

(b) Moderate Decay
Gradients Appearing

(c) Slow Decay
Gradients Ambiguity

(d) Resonate
Smooth Gradients

Figure 5: Comparison of loss landscapes: (a) Fast decay causes gradient vanishing. (b) Moderate decay improves but keeps optimal local. (c) Further slow decay makes gradient ambiguity, hindering optimization. (d) The resonance creates a smooth gradient field, aiding efficient convergence.

We further examine the loss landscape contours (Li et al., 2018) for each case in Figure 5, corresponding to Figure 4 *(right)*. Gradients are perpendicular to the contour lines, with sparse or dense contours indicating smaller or larger gradients, respectively. Extremely sparse or dense contours suggest vanishing or exploding gradients. In Figure 5 (a), **fast decay** with a small $\beta$ shows gradient vanishing, with extremely sparse contours. Increasing $\beta$ with **moderate decay** can alleviate gradient vanishing, but the model still encounters a local optima region (Figure 5 (b)). Further increasing $\beta$ with **slow decay** is expected to fully address the vanishing problem but introduces gradient ambiguity with overlapping contours (Figure 5 (c)). In contrast, the introduced oscillation from **resonate** results in smoother gradients (Figure 5 (d)), enabling more effective feature extraction.

As shown in Table 2, the PRF neuron uses only $68.9k$ parameters (768 for neurons and $67.2k$ for synapses) to achieve state-of-the-art (SOTA) results while maintaining training efficiency. Previous models required recurrent connections in the linear layers to solve the (p)s-MNIST task. To align the training parameters, we modified the architecture to a feedforward linear layer with a 1-$(128)^3$-10 structure. Using parallel computation, we completed the entire training process in only $1.60$ hours with a batch size of 256 for 200 epochs.

Table 2: The neuron for (permute)-sequential MNIST and sequential CIFAR task. The pixel-level image classification captures the hierarchical structure to verify the long-range capture ability. The symbol $*$ denotes including the **feedback linear connection**, while the symbol of $\uparrow$ and $\downarrow$ indicate that larger and smaller values are better, respectively. The BHRF result is referenced from (Higuchi et al., 2024), results of PSN family and PMSN are from (Chen et al., 2024), while other results are from (Zhang et al., 2024).

| Task (Length) | Spiking Neuron | Seq. Infer. | Par. Train. | No. Params. ↓ | Top-1 Test Acc. (%) ↑ |
|---|---|---|---|---|---|
| **sMNIST / psMNIST** (784) | LIF | ✓ | ✗ | 155.1* | 89.28 / 80.26 |
| | PLIF (Fang et al., 2021)[2021' ICCV] | ✓ | ✗ | 155.1 k* | 91.79 / – |
| | GLIF (Yao et al., 2022)[2022' NeurIPS] | ✓ | ✗ | 157.5 k* | 96.64 / 90.47 |
| | TC-LIF (Zhang et al., 2024)[2024' AAAI] | ✓ | ✗ | 155.1 k* | 99.20 / 95.36 |
| | ALIF (Yin et al., 2021)[2021' Nat. MI] | ✓ | ✗ | 156.3 k* | 98.70 / 94.30 |
| | BHRF (Higuchi et al., 2024)[2024' ICML] | ✓ | ✗ | 68.9 k | 99.10 / 95.2 |
| | **PRF (Ours)** | ✓ | ✓ | 68.9 k | **99.18 / 96.87** |
| | PSN (Fang et al., 2024)[2024' NeurIPS] | ✓ | ✓ | 2.5 M | 97.90 / 97.76 |
| | Masked PSN (Fang et al., 2024)[2024' NeurIPS] | ✓ | ✓ | 153.7 k | 97.76 / 97.53 |
| | Sliding PSN (Fang et al., 2024)[2024' NeurIPS] | ✓ | ✓ | 52.4 k | 97.20 / 82.84 |
| | PMSN (Chen et al., 2024)[2024' ArXiv] | ✓ | ✓ | 156.4 k | 99.53 / 97.78 |
| | **PRF (Ours)** | ✓ | ✓ | 167.0 k | **99.39 / 97.90** |
| **seqCIFAR** (1024) | LIF | ✓ | ✓ | 0.18 M | 45.07 |
| | PSN (Fang et al., 2024)[2024' NeurIPS] | ✓ | ✓ | 6.47 M | 55.24 |
| | Masked PSN (Fang et al., 2024)[2024' NeurIPS] | ✓ | ✓ | 0.38 M | 57.83 |
| | Sliding PSN (Fang et al., 2024)[2024' NeurIPS] | ✓ | ✓ | 0.18 M | 70.23 |
| | PMSN (Chen et al., 2024)[2024' ArXiv] | ✓ | ✓ | 0.21 M | 82.14 |
| | **PRF (Ours)** | ✓ | ✓ | 0.29 M | **82.37** |
| | **PRF (Ours)** | ✓ | ✓ | 1.10 M | **85.33** |

## 5.3 LONG RANGE ARENA TASKS

To demonstrate the long-range dependency analysis capability of our SD-TCM module, we evaluate it using the Long Range Arena (LRA) benchmark (Tay et al., 2020). This benchmark covers a wide range of classification tasks, including both textual and image domains. For the **ListOps**, **Text**, and **Retrieval** tasks, we use the **causal** architecture, while S4 employs a bidirectional architecture for all tasks. For the **Image** and **Pathfinder** tasks, we use a **bidirectional** architecture.

Table 3: Comparison of Accuracy, Parameters and Energy.

Table 4: Text (4096) ablation on $\alpha$ effective with casual architecture.

| Metric | Model | ListOps | Text | Retrieval | Image | Pathfinder | Avg. |
|---|---|---|---|---|---|---|---|
| En.(mJ) | S4 | 5.104 | 3.718 | 24.439 | 19.222 | 6.256 | 11.748 |
| | Ours | **0.075** | **0.298** | **0.211** | **0.187** | **0.067** | **0.168** |
| Acc.(%) | S4 | 59.60 | 86.82 | 90.90 | 88.65 | 94.20 | 84.03 |
| | Ours | 59.20 | 86.33 | 89.88 | 84.77 | 91.76 | 82.39 |
| Par. | S4 | 815 k | 843 k | 3.6 M | 3.6 M | 1.3 M | - |
| | Ours | 272 k | 830 k | 1.1 M | 4.1 M | 1.3 M | - |

| Text (4096) | Acc. (%) | Diff. (%) |
|---|---|---|
| without $\alpha$ | 85.75 | – |
| $+\,\alpha$ on $\mathcal{SN}$ & $\mathcal{TN}$ | 85.69 | $-\,0.06$ |
| $+\,\alpha$ on $\mathcal{TN}$ | 86.11 | $+\,0.36$ |
| $+\,\alpha$ on $\mathcal{SN}$ | **86.33** | **$+\,0.58$** |

The SD-TCM module achieves performance comparable to S4 while reducing energy consumption by over **98.57%** on average, as shown in Table 3. Specifically, the accuracy on ListOps is 59.60%, compared to S4's 59.20%. The key advantage is the significant reduction in energy consumption, as shown in Table 3, with ListOps dropping from 5.104 mJ to 0.075 mJ. This energy reduction mainly benefits from the extreme sparsity of spikes, with detailed firing rate statistics provided in Appendix M. Additionally, the model is sensitive to the imaginary part of $\theta$, causing fluctuations with different initialization (see Appendix N for details). Table 4 shows an ablation study on the $\alpha$ parameter in the Text (4096) task. Applying $\alpha$ to both the spatial ($\mathcal{SN}$) and PRF ($\mathcal{TN}$) components slightly lowers accuracy from 85.75% to 85.69%, but applying it only to $\mathcal{SN}$ improves accuracy to 86.33%, demonstrating its benefit for spatial dependencies. Finally, as shown in Table 5, our module achieves performance comparable to the S4 baseline across tasks while preserving spike-driven feature, avoiding nonlinear activation functions and FP MAC operations.

Table 5: Test Accuracy Comparison on LRA Tasks (%) ($\uparrow$). 'NL Act. Free' and 'FP MAC Free' denote models that do not use nonlinear activation functions or floating-point multiply-accumulate operations in the block, respectively. The underline and **bold** formatting indicate the SoTA result for Spikinglized SSMs and Improving Neuron methods, respectively.

| Model (Input length) | NL Act. -Free | FP MAC -Free | ListOps (2,048) | Text (4,096) | Retrieval (4,000) | Image (1,024) | Pathfinder (1,024) | Avg. |
|---|---|---|---|---|---|---|---|---|
| Random (Lower Bound) | - | - | 10.00 | 50.00 | 50.00 | 10.00 | 50.00 | 34.00 |
| Transformer (Vaswani et al., 2017) | ✗ | ✗ | 36.37 | 64.27 | 57.46 | 42.44 | 71.40 | 54.39 |
| S4 (*Bidirectional*) (Gu et al., 2022a) | ✗ | ✗ | 59.60 | 86.82 | 90.90 | 88.65 | 94.20 | 84.03 |
| Binary S4D (Stan & Rhodes, 2023) [2024' Sci. Rep.] | ✗ | ✗ | 54.80 | 82.50 | 85.03 | 82.00 | 82.60 | 77.39 |
| ↪ + GSU & GeLU | ✗ | ✗ | 59.60 | 86.50 | 90.22 | 85.00 | 91.30 | 82.52 |
| Stoch. SpikingS4 (Bal & Sengupta, 2024) [2024' arXiv] | ✗ | ✗ | 55.70 | 77.62 | 88.48 | 80.10 | 83.41 | 77.06 |
| SpikingSSMs (Shen et al., 2024) [2024' arXiv] | ✗ | ✗ | 60.23 | 80.41 | 88.77 | 88.21 | 93.51 | 82.23 |
| Spiking LMU (Liu et al., 2024) [2024' ICLR] | ✓ | ✗ | 37.30 | 65.80 | 79.76 | 55.65 | 72.68 | 62.23 |
| ELM Neuron (Spieler et al., 2024) [2024' ICLR] | ✗ | ✗ | 44.55 | 75.40 | 84.93 | 49.62 | 71.15 | 69.25 |
| **Spike-Driven TCM** | ✓ | ✓ | **59.20** | **86.33** | **89.88** | **84.77** | **91.76** | **82.39** |

## 6 CONCLUSION

This study aims to solve the SNNs problem of parallelization and performance on long sequences. We propose the decoupled reset method, enabling spiking neurons could parallel training. This method can be applied to any type of neuron to speed up. Additionally, we introduce the PRF neuron, incorporating the reset as an imaginary part to formulate oscillations in the membrane potential, which solves the long-range dependency problem. The SD-TCM model, combined with PRF neurons, achieves performance comparable to S4 on the LRA task while reducing energy consumption by two orders of magnitude. However, due to the sensitivity of training to neuron initialization, the PathX problem remains unsolved. This issue could be solved in the future by using better hyperparameters and initialization strategies.

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

## A    NOTATION IN THE PAPER

Throughout this paper and in this Appendix, we use the following notations. Matrices are represented by bold italic capital letters, such as $\boldsymbol{W}$, while sequences are represented by bold non-italic capital letters, such as $\mathbf{X}_T = \{x_1, x_2, ..., x_T\}$. For a function $f(\boldsymbol{x}) : \mathbb{R}^{d_1} \to \mathbb{R}^{d_2}$, we use $\nabla \boldsymbol{x} f$ instead of $\frac{\partial f}{\partial \boldsymbol{x}}$ to denote the first-order derivative of $f$ with respect to $\boldsymbol{x}$. The symbols $\odot$ and $\langle \cdot, \cdot \rangle$ represent the element-wise product and the inner product, respectively.

## B    THE ALGORITHM OF PSEUDO-CODE

Algorithms 1 and 2 describe the parallel computation of the LIF model, while Algorithm 3 outlines the computation process for the PRF model.

---

**Algorithm 1:** Parallel Computation of LIF

1: **Input:** $x : (T, B, N)$
2: **Output:** $y : (T, B, N) \in \{0, 1\}$
3: $K : (T) \leftarrow (\beta^0, \beta^1, \ldots, \beta^{(T-1)})$
4: $\qquad\qquad\qquad$ ▷ Expand to $(T, 1, 1)$
5: $U : (T, B, N) \leftarrow$
$\quad$ iFFT (FFT$(x) \times$ FFT$(K)$)
6: $\qquad\qquad\qquad$ ▷ Fast Fourier Transf.
7: $D : (T, B, N) \leftarrow f_D(U)$
8: $\qquad\qquad\qquad$ ▷ Scanning decoupled reset
9: $y : (T, B, N) \leftarrow \mathcal{H}(U - D)$
10: $\qquad\qquad\qquad$ ▷ Equivalent Seq. Outputs
11: **return** $y$

---

**Algorithm 2:** $f_D(U)$ decoupled reset

1: **Input:** $U : (T), V_{\text{th}} : float, \beta : float,$
$\quad T : int$
2: **Output:** $D : (T)$
3: $V : (T) \leftarrow (0, 0, \ldots, 0)$ $\quad$ ▷ Initial Empty
4: $A_{\text{bias}}, d_{\text{current}} \leftarrow 0, V_{\text{th}}$
5: **for** $t \leftarrow 0$ to $T$ **do:**
6: $\quad d[t] \leftarrow d_{\text{current}}$
7: $\quad$ **if** $U[t] \geq d_{\text{current}}$ **:**
8: $\qquad A_{\text{bias}} \leftarrow A_{\text{bias}} + 1$
9: $\quad A_{\text{bias}} \leftarrow \beta \times A_{\text{bias}}$
10: $\quad d_{\text{current}} \leftarrow V_{\text{th}} \times A_{\text{bias}} + V_{\text{th}}$
11: **return** $D$

---

**Algorithm 3:** Parallel Computation of PRF Model

1: **Input:** $x : (T, B, N), \theta : (N), \Delta : (N), \tau : float, V_{\text{th}} : float$
2: **Output:** $y : (T, B, N) \in \{0, 1\}$
3: $A : (N) \leftarrow \exp(\Delta \odot (-1/\tau + 1j \times \theta))$
4: $K : (T, N) \leftarrow (\Delta \times A^{(0)}, \Delta \times A^{(1)}, \ldots, \Delta \times A^{(T-1)})$ $\qquad$ ▷ Expand as $(T, 1, N)$ Dimension
5: $U : (T, B, N) \leftarrow$ iFFT (FFT$(x) \times$ FFT$(K)$) $\qquad$ ▷ (Inverse) Fast Fourier Transf.
6: $y : (T, B, N) \leftarrow \mathcal{H}(U.real - V_{\text{th}})$
7: **return** $y$

---

## C    THE DISCRETIZATION OF PRF NEURONS

Firs, we recall the PRF model:

$$\frac{d\tilde{u}(t)}{dt} = \tilde{\gamma}\tilde{u}(t) + c(t), \qquad (23)$$

where complex membrane potential $\tilde{u}(t) = u(t) + ir(t)$ and a complex decay constant $\tilde{\gamma} = \gamma + i\theta$ with $i = \sqrt{-1}$.

Note that here $c(t)$ is treated as a fixed external current input, which is constant from the point of view of this ODE in $u_i(t)$. Let $c_k$ denote the average value in each discrete time interval. Assumes the value of a sample of u is held constant for a duration of one sample interval $\delta$.

$$c_{t_k} = \frac{1}{\Delta t_k} \int_{t_{k-1}}^{t_k} c(t)dt \qquad (24)$$

Since we only observe the real part of the membrane potential, the input is a real number. Thus, we can consider the real part $\gamma = -\frac{1}{\tau}$ to describe the model. The model can then be expressed as:

$$\frac{du(t)}{dt} = -\frac{1}{\tau}u(t) + \frac{R}{\tau}c(t)$$

$$e^{\frac{t}{\tau}}\frac{du(t)}{dt} = -\frac{1}{\tau}e^{\frac{t}{\tau}}u(t) + e^{\frac{t}{\tau}}\frac{R}{\tau}c(t)$$

$$e^{\frac{t}{\tau}}\frac{du(t)}{dt} + \frac{1}{\tau}e^{\frac{t}{\tau}}u(t) = e^{\frac{t}{\tau}}\frac{R}{\tau}c(t)$$

$$\frac{d}{dt}\left(e^{\frac{t}{\tau}}u(t)\right) = e^{\frac{t}{\tau}}\frac{R}{\tau}c(t) \tag{25}$$

$$\int_{t_{k-1}}^{t_k}\frac{d}{dt}\left(e^{\frac{t}{\tau}}u(t)\right) = \int_{t_{k-1}}^{t_k}e^{\frac{t}{\tau}}\frac{R}{\tau}c(t)dt$$

$$e^{\frac{t_k}{\tau}}u(t_k) - e^{\frac{t_{k-1}}{\tau}}u(t_{k-1}) = \left(e^{\frac{t_k}{\tau}} - e^{\frac{t_{k-1}}{\tau}}\right)Rc_{t_k}$$

$$u(t_k) = e^{-\frac{\Delta t_k}{\tau}}u(t_{k-1}) + \left(1 - e^{-\frac{\Delta t_k}{\tau}}\right)Rc_{t_k}$$

$$u(t_k) \approx e^{-\frac{\Delta t_k}{\tau}}u(t_{k-1}) + \Delta t_k\frac{R}{\tau}c_{t_k}.$$

Rearranging, assuming $R = \tau$ without input decay, we replace $\Delta t_k$ with $\Delta$, as done in S4 (Gu et al., 2022a). We obtain the discrete form:

$$u_t = e^{\Delta\gamma}u_{t-1} + \Delta c_t. \tag{26}$$

Finally, replacing the original part $\tilde{\gamma} = -\gamma + i\theta$ gives the PRF neuron with sequential computation:

$$\tilde{u}_t = \exp\left(\Delta\left(-\frac{1}{\tau} + i\theta\right)\right)\tilde{u}_{t-1} + \Delta c_t, \tag{27}$$

$$s_t = \mathcal{H}(\Re\{\tilde{u}_t\} - V_{\text{th}}). \tag{28}$$

## D   THE FREQUENCY RESPONSE FOR PRF NEURON

This section mainly discusses the frequency response of the dynamic reset LIF neuron. Recall the membrane potential dynamic equation:

$$\tilde{u}_t = \exp\left(\Delta\left(-\frac{1}{\tau} + i\theta\right)\right)\tilde{u}_{t-1} + \Delta c_t, \tag{29}$$

$$s_t = \mathcal{H}(\Re\{\tilde{u}_t\} - V_{\text{th}}). \tag{30}$$

This dynamic process can be regarded as a damped harmonic oscillator. The real part of the membrane potential, $\Re\{u_t\}$, represents the displacement. When the displacement exceeds a certain value, this model will issue a signal. Here, $\Delta$ represents the timestep size, and $c_t$ is the input for each timestep, driven by an external force.

First, define $\tilde{\gamma} \triangleq \left(-\frac{1}{\tau} + i\theta\right)$. The model can then be expressed as:

$$\tilde{u}_t = \exp\left(\Delta\tilde{\gamma}\right)\tilde{u}_{t-1} + \Delta c_t, \tag{31}$$

$$\frac{\tilde{u}_t - \tilde{u}_{t-1}}{\Delta} = \frac{\exp\left(\Delta\tilde{\gamma}\right) - 1}{\Delta}\tilde{u}_{t-1} + c_t, \tag{32}$$

To explore the numerical effects in the experiment, we obtain the approximate ODE by using $x$ to represent $u$:

$$\frac{dx}{d\Delta} - \left(\frac{\exp\left(\Delta\tilde{\gamma}\right) - 1}{\Delta}\right)x = c_t, \tag{33}$$

using a first-order Taylor expansion approximation, we obtain:

$$\frac{dx}{d\Delta} - \tilde{\gamma}x = c_t, \tag{34}$$

We assume the driven input is an oscillation, $c_t = c_0 \exp(i\omega\Delta)$, with a constant base amplitude $c_0$ and a variable angular frequency $\omega$. The position of the membrane potential $x$ will oscillate in resonance as:

$$x = x_\omega \exp(i\omega\Delta), \tag{35}$$

where $x_\omega$ is the amplitude as a function of the external excitation frequency. We then have:

$$\dot{x} = i\omega x_\omega \exp(i\omega\Delta). \tag{36}$$

Substituting Equation 36 into Equation 34 gives:

$$i\omega x_\omega \exp(i\omega\Delta) - \tilde{\gamma} x_\omega \exp(i\omega\Delta) = c_0 \exp(i\omega\Delta) \tag{37}$$

$$i\omega x_\omega - \tilde{\gamma} x_\omega = c_0 \tag{38}$$

$$(i\omega - \tilde{\gamma}) x_\omega = c_0 \tag{39}$$

Rearranging the equation yields:

$$\frac{x_\omega}{c_0} = \frac{1}{i\omega - \tilde{\gamma}} \tag{40}$$

$$= \frac{1}{\frac{1}{\tau} + i(\omega - \theta)}. \tag{41}$$

Thus, we have:

$$\Re\left\{\frac{x_\omega}{c_0}\right\} = \frac{1/\tau}{(\frac{1}{\tau})^2 + (\omega - \theta)^2} \tag{42}$$

$$\Im\left\{\frac{x_\omega}{c_0}\right\} = \frac{-\omega + \theta}{(\frac{1}{\tau})^2 + (\omega - \theta)^2} \tag{43}$$

The magnitude can be obtained by taking the modulus, which varies with $\omega$ and $\theta$:

$$\left|\frac{x_\omega}{c_0}\right| = \sqrt{\Re\left\{\frac{x_\omega}{c_0}\right\}^2 + \Im\left\{\frac{x_\omega}{c_0}\right\}^2} \tag{44}$$

$$= \frac{1}{\sqrt{(\frac{1}{\tau})^2 + (\omega - \theta)^2}} \tag{45}$$

Assuming $\omega > 0$, the value of $\omega$ corresponding to the maximum of the magnitude can be found:

$$d\left|\frac{x_\omega}{c_0}\right|/d\omega = 0 \tag{46}$$

$$-\frac{1}{2}\left((\frac{1}{\tau})^2 + (\omega - \theta)^2\right)^{-\frac{3}{2}} \times 2(\omega - \theta) = 0 \tag{47}$$

$$\omega = \theta \tag{48}$$

Therefore, the resonant frequency $\omega$ at the point of maximum magnitude coincides with $\theta$, and $\max\left(\left|\frac{x_\omega}{c_0}\right|\right) = \tau$.

## E    DEPLOYMENT ANALYSIS OF PRF NEURON

This model can also be easily deployed on neuromorphic chips. After training, the coefficients can merge together.

Recalling the PRF Neuron as described in Equation 19, we explicitly expand the real and imaginary parts as follows:

$$\tilde{u}_t = \exp\left(\Delta\left(-\frac{1}{\tau} + i\theta\right)\right)\tilde{u}_{t-1} + \Delta c_t$$

$$= \left(\exp\left(-\frac{\Delta}{\tau}\right)\cos(\Delta\theta) + i\exp\left(-\frac{\Delta}{\tau}\right)\sin(\Delta\theta)\right)\tilde{u}_{t-1} + \Delta c_t \tag{49}$$

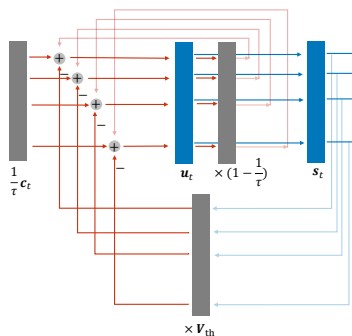 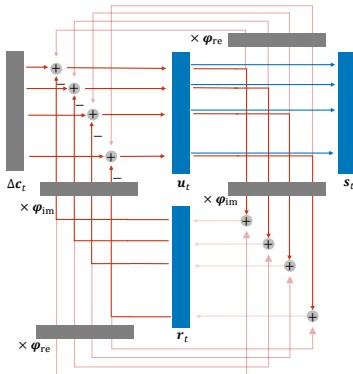

Figure 6: The comparison of LIF Neuron (*left*) and PRF Neuron (*right*) for inference deployment.

We use the symbols $\Re\tilde{u}_t \in \mathbb{R}$ and $\Im\tilde{u}_t \in \mathbb{R}$ to denote the real and imaginary parts of the membrane potential $\tilde{u}_t \in \mathbb{C}$, respectively:

$$\begin{pmatrix} \Re\{\tilde{u}_t\} \\ \Im\{\tilde{u}_t\} \end{pmatrix} = \begin{pmatrix} \varphi_{\mathrm{re}} & -\varphi_{\mathrm{im}} \\ \varphi_{\mathrm{im}} & \varphi_{\mathrm{re}} \end{pmatrix} \begin{pmatrix} \Re\{\tilde{u}_{t-1}\} \\ \Im\{\tilde{u}_{t-1}\} \end{pmatrix} + \begin{pmatrix} \Delta c_t \\ 0 \end{pmatrix}, \tag{50}$$

where the coefficients $\varphi_{\mathrm{re}}, \varphi_{\mathrm{im}} \in \mathbb{R}$ are the merged parameters:

$$\varphi_{\mathrm{re}} = \exp\left(-\frac{\Delta}{\tau}\right)\cos\left(\Delta\theta\right),$$

$$\varphi_{\mathrm{im}} = \exp\left(-\frac{\Delta}{\tau}\right)\sin\left(\Delta\theta\right). \tag{51}$$

Finally, the spike is output based on the real part $\Re\tilde{u}_t$. For simplicity, we use $u_t$ and $r_t$ to denote $\Re\tilde{u}_t$ and $\Im\tilde{u}_t$, respectively. Thus, the explicit iteration of the PRF can be written as:

$$u_t = \varphi_{\mathrm{re}}u_{t-1} - \varphi_{\mathrm{im}}r_{t-1} + \Delta c_t, \tag{52}$$
$$r_t = \varphi_{\mathrm{im}}u_{t-1} + \varphi_{\mathrm{re}}r_{t-1}, \tag{53}$$
$$s_t = \mathcal{H}\left(u_t - V_{\mathrm{th}}\right). \tag{54}$$

Intuitively, compared to the LIF model, the PRF introduces two additional multiplication operations and one extra addition operation for inference, along with an extra hidden state that needs to be saved, as shown in Figure 6. Furthermore, this neuron model, with its double hidden state, can also be easily deployed on neuromorphic chips, similar to how the AHP model (Rao et al., 2022) was deployed on the Loihi chip (Davies et al., 2018).

## F PROOF

### F.1 PROOF OF THEOREM.1

*Proof.* Firstly, consider the Adaptive LIF (ALIF) model (Bellec et al., 2020), where the threshold adapts according to recent firing activity. The dynamic threshold is given by Equation 55 to Equation 57:

$$A_t = V_{\mathrm{th}} + \beta a_t \tag{55}$$
$$z_t = \mathcal{H}(u_t - A_t) \tag{56}$$
$$a_{t+1} = \rho a_t + z_t \tag{57}$$

Here, the decay factor $\rho$ is given by $e^{-\delta t/\tau_a}$, where $\tau_a$ is the adaptation time constant, as described in ALIF (Bellec et al., 2020).

Intuitively, when $\tau_a \gg \delta t$, we have $\rho \to 1$, simplifying Equation 57 to $a_{t+1} = a_t + z_t$. The variable $z_t$ can be further deduced as follows:

$$z_t = \mathcal{H}(u_t - A_t) = \begin{cases} 0, & \text{if } u_t < A_t \\ 1, & \text{if } u_t \geq A_t \end{cases} \tag{58}$$

Substituting Equation 58 into Equation 57, we obtain:

$$a_{t+1} = \begin{cases} a_t, & \text{if } u_t < A_t \\ a_t + 1, & \text{if } u_t \geq A_t \end{cases} \tag{59}$$

Therefore, Equation 55 can be further expanded using Equation 59:

$$A_t = V_{\text{th}} + a_t \tag{60}$$

$$a_t = \begin{cases} \beta a_{t-1}, & \text{if } u_t < A_t \\ \beta(a_{t-1} + 1), & \text{if } u_t \geq A_t \end{cases} \tag{61}$$

Finally, we can see that Equation 60 - Equation 61 are equivalent to Equation 10 - Equation 11. $\square$

### F.2 PROOF OF THEOREM.2

*Proof.* Firstly, we recall the dynamic iteration of the PRF model:

$$\tilde{u}_t = \exp\left(\Delta\left(-\frac{1}{\tau} + i\theta\right)\right)\tilde{u}_{t-1} + \Delta c_t, \tag{62}$$

$$s_t = \mathcal{H}\left(\Re\{\tilde{u}_t\} - V_{\text{th}}\right) \tag{63}$$

Next, let $\Delta = 1$ and $\theta = 0$, allowing the model to be rewritten as:

$$u_t = \exp\left(-\frac{1}{\tau}\right)u_{t-1} + c_t, \tag{64}$$

$$s_t = \mathcal{H}\left(u_t - V_{\text{th}}\right) \tag{65}$$

At this point, $u_t \in \mathbb{R}$, meaning it only contains a real part, with the decay affecting only the real component. Setting $\theta = 0$ can be interpreted as removing the reset process. Furthermore, the exponential term $\exp\left(-\frac{1}{\tau}\right)$ can be approximated using the first-order Taylor expansion:

$$\exp\left(-\frac{1}{\tau}\right) \approx 1 - \frac{1}{\tau} \tag{66}$$

Finally, the dynamic equation can be rewritten as:

$$u_t = \left(1 - \frac{1}{\tau}\right)u_{t-1} + c_t, \tag{67}$$

$$s_t = \mathcal{H}\left(u_t - V_{\text{th}}\right) \tag{68}$$

Equation 67 and Equation 68 are equivalent to the standard LIF model without the reset process, as given in Equation 2. $\square$

### F.3 PROOF OF THEOREM.3

*Proof.* Firstly, we recall the dynamic iteration of the PRF model:

$$\tilde{u}_t = \exp\left(\Delta\left(-\frac{1}{\tau} + i\theta\right)\right)\tilde{u}_{t-1} + \Delta c_t, \tag{69}$$

assuming $c_t \sim \mathcal{N}(0, \sigma^2)$ is normally distributed with zero mean and variance $\sigma^2$ at time-invariant, and $\Delta$, $\tau$, and $\theta$ are constants. Define the complex constant $A$:

$$A = \exp\left(\Delta\left(-\frac{1}{\tau} + i\theta\right)\right) = \exp\left(-\frac{\Delta}{\tau}\right)\exp\left(i\Delta\theta\right). \tag{70}$$

Note that the magnitude of $A$ is:

$$|A| = e^{-\Delta/\tau} < 1, \tag{71}$$

since $\tau > \Delta > 0$. In further, where the membrane potential $u_t = \Re\{\tilde{u}_t\}$ is the real part of $\tilde{u}_t$.

Next, we expand $\tilde{u}_t$ recursively:

$$\tilde{u}_t = A\tilde{u}_{t-1} + \Delta c_t \tag{72}$$
$$= A(A\tilde{u}_{t-2} + \Delta c_{t-1}) + \Delta c_t \tag{73}$$
$$= A^2\tilde{u}_{t-2} + A\Delta c_{t-1} + \Delta c_t \tag{74}$$
$$\vdots \tag{75}$$
$$= A^t\tilde{u}_0 + \Delta\sum_{k=1}^{t} A^{t-k}c_k. \tag{76}$$

Now, we calculate the expectation and variation of $u_t$. since $c_t$ are independent and identically distributed with $\mathbb{E}[c_t] = 0$ and $\mathrm{Var}(c_t) = \sigma^2$, we can compute the expected value of $\tilde{u}_t$:

$$\mathbb{E}[\tilde{u}_t] = \mathbb{E}\left[A^t\tilde{u}_0 + \Delta\sum_{k=1}^{t} A^{t-k}c_k\right] \tag{77}$$

$$= A^t\tilde{u}_0 + \Delta\sum_{k=1}^{t} A^{t-k}\mathbb{E}[c_k] \tag{78}$$

$$= A^t\tilde{u}_0. \tag{79}$$

As $|A| < 1$, it follows that:

$$\lim_{t\to\infty}\mathbb{E}[u_t] = \Re\{\lim_{t\to\infty}\mathbb{E}[\tilde{u}_t]\} \to 0 \tag{80}$$

Next, compute the variance of $\tilde{u}_t$ :

$$\mathrm{Var}(\tilde{u}_t) = \mathbb{E}\left[|\tilde{u}_t|^2\right] - |\mathbb{E}[\tilde{u}_t]|^2 \tag{81}$$

$$= \mathbb{E}\left[\left|A^t\tilde{u}_0 + \Delta\sum_{k=1}^{t} A^{t-k}c_k\right|^2\right] - |A^t\tilde{u}_0|^2 \tag{82}$$

$$= 2A^t\tilde{u}_0\Delta\sum_{k=1}^{t} A^{t-k}\mathbb{E}[c_k] + \Delta^2\mathbb{E}\left[\left|\sum_{k=1}^{t} A^{t-k}\mathbb{E}[c_k]\right|^2\right] \tag{83}$$

$$= \Delta^2\mathbb{E}\left[\left|\sum_{k=1}^{t} A^{t-k}c_k\right|^2\right]. \tag{84}$$

Since $c_k$ are independent and have zero mean, we have:

$$\mathbb{E}\left[\left|\sum_{k=1}^{t} A^{t-k} c_k\right|^2\right] = \sum_{k=1}^{t}\sum_{l=1}^{t} A^{t-k}\overline{A^{t-l}}\mathbb{E}\left[c_k c_l\right] \tag{85}$$

$$= \sum_{k=1}^{t} |A|^{2(t-k)}\mathbb{E}\left[|c_k|^2\right], \quad \text{since } \mathbb{E}\left[c_k c_l\right] = 0 \text{ for } k \neq l \tag{86}$$

$$= \sigma^2 \sum_{k=1}^{t} |A|^{2(t-k)}. \tag{87}$$

Therefore, the variance of $u_t$ is:

$$\mathrm{Var}(u_t) = \Delta^2 \sigma^2 \sum_{k=1}^{t} |A|^{2(t-k)} = \Delta^2 \sigma^2 \sum_{m=0}^{t-1} |A|^{2m}. \tag{88}$$

Since $|A| = e^{-\Delta/\tau}$ as deduced in Equation 71, we have:

$$|A|^{2m} = e^{-2\Delta m/\tau}. \tag{89}$$

Thus,

$$\mathrm{Var}(u_t) = \Delta^2 \sigma^2 \sum_{m=0}^{t-1} e^{-2\Delta m/\tau}. \tag{90}$$

This is a finite geometric series with first term equals to $1$ and common ratio $r = e^{-2\Delta/\tau}$:

$$\sum_{m=0}^{t-1} e^{-2\Delta m/\tau} = \frac{1-r^t}{1-r}. \tag{91}$$

Therefore,

$$\mathrm{Var}(u_t) = \Delta^2 \sigma^2 \frac{1 - e^{-2\Delta t/\tau}}{1 - e^{-2\Delta/\tau}}. \tag{92}$$

As $t \to \infty$, $e^{-2\Delta t/\tau} \to 0$, so the variance approaches:

$$\lim_{t\to\infty} \mathrm{Var}(u_t) = \frac{1}{1 - e^{-2\Delta/\tau}} \approx \frac{\Delta^2 \sigma^2}{2\Delta/\tau} = \frac{\tau\Delta}{2}\sigma^2, \tag{93}$$

this is a finite constant, in summary we could get the distribution after $t \to \infty$:

$$u_t \sim \mathcal{N}\left(0, \frac{\tau\Delta}{2}\sigma^2\right), \tag{94}$$

which implies that $u_t$ is bounded in probability. This indicates that $u_t$ does not diverge but instead stabilizes to a specific distribution related to the input distribution and hyperparameters. It ensures that despite the randomness introduced by the inputs $c_t$, the neuron's response remains predictable in distribution. $\qquad\square$

## G  PARALLEL PERSPECTIVE ON SOLVING LONG-RANGE LEARNING PROBLEM

From the subsection of Problem Formulation 2, we gain the insight that the gradient is proportional to the inner product of the kernel and previous layer spikes.

$$\nabla_{\boldsymbol{W}^l}\mathcal{L} \propto \frac{\partial \mathcal{L}}{\partial u_T^l} \sum_{t=1}^{T} \frac{\partial u_T^l}{\partial u_t^l} \frac{\partial u_t^l}{\partial \boldsymbol{W}^l} \propto \underbrace{\sum_{t=1}^{T} \beta^{(t)} s_t^{l-1}}_{\text{Parallel Perspective}} = \left\langle \boldsymbol{K}_T, \boldsymbol{S}_{T:1}^{l-1}\right\rangle \tag{95}$$

To verify this insight, we define three extreme situations (**Fast Decay**, **Slow Decay** and **Slow Decay with Resonate**) to explore this from the parallel kernel perspective, as shown in Figure 7.

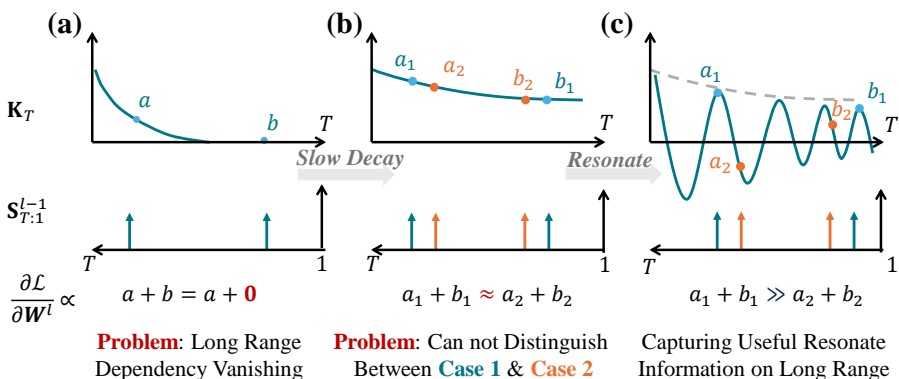

Figure 7: The parallel kernel perspective for learning long-range abilities.

**Fast Decay**  Fast decay may cause the problem of long-range dependency vanishing, as shown in Figure 7(a). The kernel $\mathbf{K}_T$ decays rapidly as the time step $T$ increases. In this case, early spikes (represented by $a$) dominate the gradient calculation, while contributions from later spikes (represented by $b$) are almost negligible. As a result, the gradient $\partial \mathcal{L}/\partial \boldsymbol{W}^l$ is mostly proportional to $a$, leading to the vanishing of long-range dependencies. The network struggles to learn and retain information from distant time steps, causing the gradients to vanish and impairing the learning of long-term dependencies.

**Slow Decay**  Slow Decay may relive the vanishing problem, but may cause gradients ambiguity as shown in Figure.7(b). This situation shows the kernel decays more slowly over time, which balances the contributions from both early and late spikes (denoted as $a_1, b_1$ and $a_2, b_2$). However, this slow decay introduces a new problem—ambiguity. When the contributions from different parts of the sequence are similar (e.g., $a_1 + b_1 \approx a_2 + b_2$), the network finds it difficult to distinguish between these cases. This ambiguity can confuse the learning process, as the network may not correctly interpret or differentiate between distinct temporal patterns.

**Slow Decay with Resonate**  Resonate could may relive the ambiguity problem to capture the resonate information under the long range, as shown in Figure.7(c). the kernel $\mathbf{K}_T$ oscillates or resonates, effectively capturing contributions from both early and late spikes. This resonance allows the network to amplify and preserve significant information across the entire sequence, represented by $a_1$ and $b_1$. Such behavior is advantageous for learning long-range dependencies, as it helps maintain the gradient information over time. The network can now better distinguish between different temporal patterns, leading to improved learning and retention of long-term information, which is crucial for tasks involving long sequences.

## H    THE ARCHITECTURE FOR LONG RANGE ARENA TASK

This Section introduces the *Spike-Driven Temporal and Channel Mixing (SD-TCM)* Module, as shown in Figure 8. This design philosophy mainly stems from the token and channel mixing in the transformer and S4 for solving more difficult sequence tasks. Firstly, we introduce the spatial neuron by considering the other side of LIF neurons. Secondly, we present the mixer module, which combines the PRF neuron and spatial neuron. Furthermore, we compare the computation complexity and theoretical energy consumption (Detail in Appendix.I).

The design philosophy of this module stems from combining token mixing with channel mixing, a common practice in transformer and S4 modules. The transformer uses a self-attention block for token mixing and a 2-layer MLP for channel mixing (Vaswani et al., 2017). The S4 employs SSMs for token mixing and GLU for channel mixing (Gu et al., 2022a). Firstly, we propose the Spatial neuron utilized for channel mixing. We consider the limitation of the time constant $\tau$ close to the

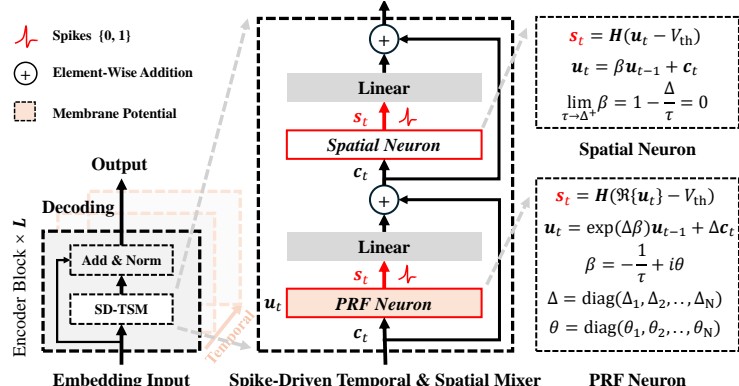

Figure 8: Diagram of the *Spike-Driven Temporal and Channel Mixer (SD-TCM)* Module.

time scale, focusing on the instantaneous information for each timestep:

$$\lim_{\tau \to 1^+} u_t = \left(1 - \frac{1}{\tau}\right) u_{t-1} + c_t \approx c_t, \tag{96}$$

then the output of Spatial Neuron replaced as $s_t = \mathcal{H}(c_t - V_{\text{th}})$. This type of neuron can also be observed in motor neurons in biology with extreme huge decay. According to Theorem 2, we gain the insight that the LIF neuron is a subset of the PRF Neuron. The Spatial Neuron is also a subset of the LIF Neuron, aiming to focus on instantaneous spatial information without temporal information. Consequently, we derived a special case of the LIF neuron, which we term the Spatial Neuron. In further, we use the trainable amplitude for Spatial LIF Neuron with the output $\{0, \alpha\}$. Where the amplitude constant $\alpha \in \mathbb{R}^+$ is trainable parameters with always initialize as 1. After training, the amplitude constant $\alpha$ could merge to the following *Linear* layer during the inference. That means $(\alpha s_t) \times W \equiv s_t \times (\alpha W)$.

Secondly, we design the SD-TCM module consists of three main components: the Spatial Neuron $\mathcal{SN}(\cdot)$, PRF Neuron $\mathcal{TN}(\cdot)$, fully-connected layer $\text{Linear}(\cdot)$. To keep the spike-driven feature, we use the membrane shortcut residual connect (Hu et al., 2024) like spike-driven transformer (Yao et al., 2024). Given a input sequence $I \in \mathbb{R}^{T \times N \times D_{\text{in}}}$, the embedding the sequence of $N$ flattened spike patches with $D$ dimensional channel,

$$\mathbf{U}_T^1 = \text{Embedding}(I), \qquad I \in \mathbb{R}^{T \times N \times D_{\text{in}}}, \quad \mathbf{U}_T \in \mathbb{R}^{T \times N \times D}, \tag{97}$$

where $T$ denote timestep while aligning with the sequence length. Then the SD-TCM is written as:

$$\mathbf{S}_T^l = \mathcal{TN}(\mathbf{U}_T^l), \qquad\qquad \mathbf{S}_T^l \in \mathbb{B}^{T \times N \times D}, \quad l = 1, 2, \ldots, L \tag{98}$$

$$\text{RPE} = \mathbf{U}_T^l + \text{Linear}(\mathbf{S}_T^l), \qquad \text{RPE} \in \mathbb{R}^{T \times N \times D}, \quad l = 1, 2, \ldots, L \tag{99}$$

$$\mathbf{S'}_T^l = \mathcal{SN}(\text{RPE}), \qquad\qquad \mathbf{S'}_T^l \in \mathbb{B}^{T \times N \times D}, \quad l = 1, 2, \ldots, L \tag{100}$$

$$\mathbf{U}_T^{l+1} = \text{RPE} + \text{Linear}(\mathbf{S'}_T^l), \qquad \mathbf{U}_T^{l+1} \in \mathbb{R}^{T \times N \times D}, \quad l = 1, 2, \ldots, L \tag{101}$$

Where $\mathbb{B} \triangleq \{0, 1\}$ is the binary value set. After the $L^{\text{th}}$ layer, the following output as the input of classifier or other head for corresponding task. This block keep the spike-driven with two properties: event-driven and binary spike-based communication. The former means that no computation is triggered when the input is zero. The binary restriction in the latter indicates that there are only additions.

The original S4 layer is unidirectional or causal, which is an unnecessary constraint for the classification tasks appearing in LRA. (Goel et al., 2022a) propose a bidirectional version of S4 that simply concatenates two S4 convolution kernels back-to-back (Gu et al., 2022b). As same the bidirectional model implemented in S4 block. We implement the bidirectional model by replacing the Equation 99 as the following equation:

$$\mathbf{S}_T^l = \textbf{Concat}\left(\mathcal{TN}(\mathbf{U}_T^l), \textbf{Rev}\left(\mathcal{TN}(\textbf{Rev}(\mathbf{U}_T^l))\right)\right), \qquad \mathbf{S}_T^l \in \mathbb{B}^{T \times N \times 2D}, \tag{102}$$

where **Concat** and **Rev** means the concatenate and reverse operation along the channel and timestep dimension in respectively. We simply pass the input sequence through an PRF Neuron, and also reverse it and pass it through an independent second PRF Neuron. These spiking outputs are concatenated and passed through a position wise linear layer. Keep the same with S4 and Shashimi (Goel et al., 2022a), the following Linear layer ($W \in \mathbb{R}^{2D \times D}$) will change the input feature dimension as the double in the bidirectional model.

# I   THE THEORETICAL ANALYSIS OF POWER CONSUMPTION

The two tables provide a detailed comparison of inference complexity and energy consumption across various models. Table 6 compares the computational complexity involved in the inference phase for different models. Table 7 evaluates the energy consumption of these models during the token mixing and channel mixing stages.

Table 6: The comparison for inference complexity. The abstract formulations $x_t$, $u_t$, and $y_t$ represent the input, hidden state, and output, respectively. The symbols $\mathbb{R}$ and $\mathbb{C}$ denote real and complex number sets. Where the symbols $\mathcal{H}$ and $\Re$ denote the Heaviside function and the real part of the complex number. The $fr$ means the firing rate $\in (0, 1)$.

| Token Mixing | Dynamic Equation | Variables | Parameters | Infer. Complexity |
|---|---|---|---|---|
| **SSMs** | $u_t = Au_{t-1} + Bx_t$ 
 $y_t = Cu_t$ | $x_t \in \mathbb{R}^D$ 
 $u_t \in \mathbb{R}^H$ 
 $y_t \in \mathbb{R}^D$ | $A \in \mathbb{R}^{H \times H}$ 
 $B \in \mathbb{R}^{D \times H}$ 
 $C \in \mathbb{R}^{H \times D}$ | $O(H^2 + 2DH)$ |
| **Spikinglized SSMs** | $u_t = Au_{t-1} + Bx_t$ 
 $y_t = \mathcal{H}(Cu_t - V_{\text{th}})$ | $x_t \in \mathbb{R}^D$ 
 $u_t \in \mathbb{R}^H$ 
 $y_t \in \{0,1\}^D$ | $A \in \mathbb{R}^{H \times H}$ 
 $B \in \mathbb{R}^{D \times H}$ 
 $C \in \mathbb{R}^{H \times D}$ | $O(H^2 + 2DH)$ |
| **PRF + Linear** | $u_t = A \odot u_{t-1} + B \odot x_t$ 
 $s_t = \mathcal{H}(\Re\{u_t\} - V_{\text{th}})$ 
 $y_t = \text{Linear}(s_t)$ | $x_t \in \mathbb{R}^D$ 
 $u_t \in \mathbb{C}^D$ 
 $s_t \in \{0,1\}^D$ 
 $y_t \in \mathbb{R}^D$ | $A \in \mathbb{C}^D$ 
 $B \in \mathbb{R}^D$ 
 $W \in \mathbb{R}^{D \times D}$ | $O(5D + fr \cdot D^2)$ |

Table 7: Energy evaluation. $R$ denote the spike firing rates (the proportion of non-zero elements in the neuron output). ($\sigma$: Sigmoid activation function, $\mathcal{H}$: Heaviside function and $\Re$: real part of the complex number, and Ter means the ternary output $\{-1, 0, 1\}$ with a dynamic threshold).

| | | Token Mixing | | | Channel Mixing | | |
|---|---|---|---|---|---|---|---|
| | Comput. | Complexity | Energy | Comput. | Complexity | Energy |
| **S4-LegS** | $u_t = Au_{t-1} + Bx_t$ 
 $y_t = Cu_t + Dx_t$ | $O(H^2 + DH)$ 
 $O(HD + D^2)$ | $E_{\text{MAC}} \cdot (H^2 + DH)$ 
 $E_{\text{MAC}} \cdot (HD + D^2)$ | $n_t = \text{Linear}_1(y_t)$ 
 $m_t = \text{Linear}_2(y_t)$ 
 $o_t = n_t \odot \sigma(m_t)$ | $O(D^2)$ 
 $O(D^2)$ 
 $O(3D)$ | $E_{\text{MAC}} \cdot D^2$ 
 $E_{\text{MAC}} \cdot D^2$ 
 $E_{\text{MAC}} \cdot 2D + E_{\text{M}} \cdot D$ |
| **Binary S4D** | $u_t = Au_{t-1} + Bx_t$ 
 $y_t = \mathcal{H}(Cu_t + Dx_t)$ | $O(H^2 + DH)$ 
 $O(HD + D^2)$ | $E_{\text{MAC}} \cdot (H^2 + DH)$ 
 $E_{\text{MAC}} \cdot (HD + D^2)$ | $n_t = \text{Linear}_1(y_t)$ 
 $m_t = \text{Linear}_2(y_t)$ 
 $o_t = n_t \odot \sigma(m_t)$ | $O(R \cdot D^2)$ 
 $O(R \cdot D^2)$ 
 $O(3D)$ | $E_{\text{AC}} \cdot R \cdot D^2$ 
 $E_{\text{AC}} \cdot R \cdot D^2$ 
 $E_{\text{MAC}} \cdot 2D + E_{\text{M}} \cdot D$ |
| **GSU** | $u_t = Au_{t-1} + Bx_t$ 
 $y_t = Cu_t + Dx_t$ | $O(H^2 + DH)$ 
 $O(HD + D^2)$ | $E_{\text{MAC}} \cdot (H^2 + DH)$ 
 $E_{\text{MAC}} \cdot (HD + D^2)$ | $n_t = \text{Ter}(W_1)y_t + b_1$ 
 $m_t = W_2\text{Ter}(y_t) + b_2$ 
 $o_t = \text{GeLU}(n_t \odot m_t)$ | $O(R \cdot D^2)$ 
 $O(R \cdot D^2)$ 
 $O(3D)$ | $E_{\text{AC}} \cdot R \cdot D^2$ 
 $E_{\text{AC}} \cdot R \cdot D^2$ 
 $E_{\text{MAC}} \cdot 2D + E_{\text{M}} \cdot D$ |
| **Ours** | $u_t = A \odot u_{t-1} + B \odot x_t$ 
 $y_t = \mathcal{H}(\Re\{u_t\} - V_{\text{th}})$ 
 $n_t = \text{Linear}_1(y_t) + u_t$ | $O(5D)$ 
 - 
 $O(R \cdot D^2 + D)$ | $E_{\text{M}} \cdot 5D + E_{\text{AC}} \cdot 3D$ 
 - 
 $E_{\text{AC}} \cdot (R \cdot D^2 + D)$ | $s_t = \mathcal{H}(n_t - V_{\text{th}})$ 
 $o_t = \text{Linear}_2(s_t) + n_t$ | - 
 $O(R \cdot D^2 + D)$ | - 
 $E_{\text{AC}} \cdot (R \cdot D^2 + D)$ |

# J   DESCRIPTION OF DATASETS AND HYPERPARAMETERS

All our experiments were conducted on an NVIDIA GeForce RTX 4090 GPU with 24 GB of memory. The specific experimental setup and hyperparameters are detailed in subsection J.1, and the description of the experimental dataset is provided in subsection J.2.

## J.1 Task Specific Hyperparameters

Here we specify any task-specific details, hyperparameter or architectural differences from the defaults outlined above.

### J.1.1 Sequential MNIST & Permuted Sequential MNIST

For Figure.4, we used a neural network architecture with layers of size 1-64-256-256-10 ($87k$ training parameters) with 256 batch size for training 200 epochs. All experiments were conducted using the same random seeds.

### J.1.2 Long Range Arena

The total hyperparameters configure is shown in Table 8.

Table 8: Hyperparameters for LRA Task

| Task | Depth | Channels | Norm | Pre-norm | Dropout | LR | Neuron LR | B | Epochs | WD | $(\Delta_{min}, \Delta_{max})$ |
|------|-------|----------|------|----------|---------|-----|-----------|---|--------|-----|-------------------------------|
| **ListOps** | 8 | 128 | BN | False | 0 | 0.005 | 0.001 | 50 | 40 | 0.05 | (0.001, 0.1) |
| **Text** | 6 | 256 | BN | True | 0 | 0.005 | 0.001 | 16 | 32 | 0.05 | (0.001, 0.1) |
| **Retrieval** | 6 | 256 | BN | True | 0 | 0.005 | 0.001 | 32 | 20 | 0.05 | (0.001, 0.1) |
| **Image** | 6 | 512 | BN | False | 0.1 | 0.005 | 0.001 | 50 | 200 | 0.05 | (0.001, 0.1) |
| **Pathfinder** | 6 | 256 | BN | True | 0.05 | 0.005 | 0.001 | 64 | 200 | 0.03 | (0.001, 0.1) |

## J.2 Dataset Details

We provide more context and details for (p)s-MNIST and each tasks of the LRA (Tay et al., 2021). Note that we follow the same data pre-processing steps as (Gu et al., 2022a), which we also include here for completeness. The following describe mainly refer from (Smith et al., 2023).

- **Sequential MNIST**: (sMNIST) 10-way digit classification from a $28 \times 28$ grayscale image of a handwritten digit, where the input image is flattened into a $784$-length scalar sequence.

- **Permuted Sequential MNIST**: (psMNIST) 10-way digit classification from a $28 \times 28$ grayscale image of a handwritten digit, where the input image is flattened into a $784$-length scalar sequence. This sequence is then permuted using a fixed order.

- **ListOps**: A lengthened version of the dataset presented by (Nangia & Bowman, 2018). Given a nested set of mathematical operations (such as **min** and **max**) and integer operands in the range zero to nine, expressed in prefix notation with brackets, compute the integer result of the mathematical expression (e.g. [**max** 2 6 [ **min** 9 7 ] 0] $\rightarrow$ 7). Characters are encoded as one-hot vectors, with 17 unique values possible (opening brackets and operators are grouped into a single token). The sequences are of unequal length, and hence the end of shorter sequences is padded with a fixed indicator value, padded to a maximum length of $2,000$. A reserved end-of-sequence token is appended. There are 10 different classes, representing the integer result of the expression. There are 96, 000 training sequences, 2, 000 validation sequences, and 2, 000 test sequences. No normalization is applied.

- **Text**: Based off of the iMDB sentiment dataset presented by (Maas et al., 2011). Given a movie review, where characters are encoded as a sequence of integer tokens, classify whether the movie review is positive or negative. Characters are encoded as one-hot vectors, with 129 unique values possible. Sequences are of unequal length, and are padded to a maximum length of $4,096$. There are two different classes, representing positive and negative sentiment. There are 25, 000 training examples and 25, 000 test examples. No validation set is provided. No normalization is applied.

- **Retrieval**: Based off of the ACL Anthology network corpus presented by (Radev et al., 2009). Given two textual citations, where characters are encoded as a sequence of integer tokens, classify whether the two citations are equivalent. The citations must be compressed separately, before being passed into a final classifier layer. This is to evaluate how effectively the network can represent the text. The decoder head then uses the encoded representation to complete the task. Characters are encoded into a one-hot vector with 97 unique values.

Two paired sequences may be of unequal length, with a maximum sequence length of $4,000$. There are two different classes, representing whether the citations are equivalent or not. There are $147,086$ training pairs, $18,090$ validation pairs, and $17,437$ test pairs. No normalization is applied.

- **Image**: Uses the CIFAR-10 dataset presented by (Krizhevsky, 2009). Given a $32 \times 32$ grayscale CIFAR-10 image as a one-dimensional raster scan, classify the image into one of ten classes. Sequences are of equal length $(1,024)$. There are ten different classes. There are $45,000$ training examples, $5,000$ validation examples, and $10,000$ test examples. RGB pixel values are converted to a grayscale intensities, which are then normalized to have zero mean and unit variance (across the entire dataset).

- **Pathfinder**: Based off of the Pathfinder challenge introduced by (Linsley et al., 2018). A $32 \times 32$ grayscale image image shows a start and an end point as a small circle. There are a number of dashed lines on the image. The task is to classify whether there is a dashed line (or path) joining the start and end point. There are two different classes, indicating whether there is a valid path or not. Sequences are all of the same length $(1,024)$. There are $160,000$ training examples, $20,000$ validation examples, and $20,000$ test examples. The data is normalized to be in the range $[-1, 1]$.

## K  THE EQUIVALENCE OF SEQUENTIAL AND PARALLEL

First, the parallel computation is equivalent to sequential computation during both inference and training phases. This equivalence is clearly illustrated in Figures 9 and 10. For inference equivalence, Figure 9 shows that when applying random input to an LIF neuron using both sequential and parallel computation, the spiking output remains consistent across both methods.

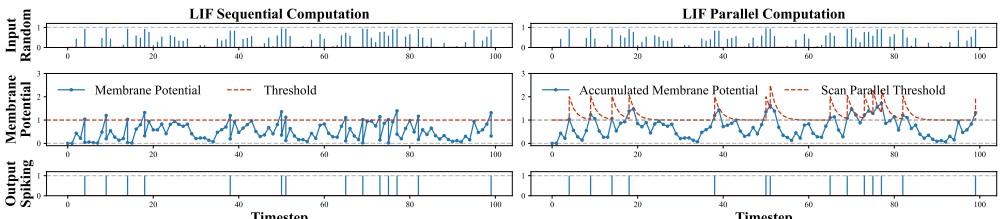

Figure 9: **Verification of inference equivalence** for parallel and sequential computation. With random input, the spiking output from parallel computation is equivalent to that of sequential computation.

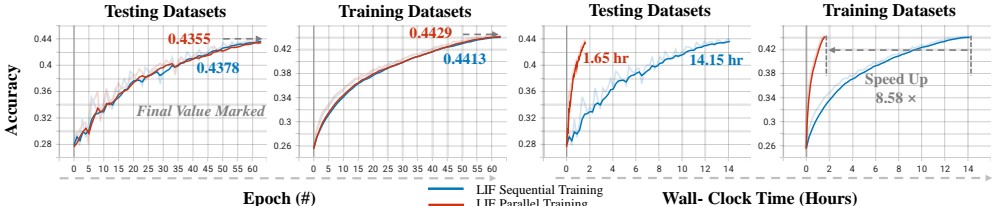

Figure 10: **Verification of training equivalence** between sequential and parallel training. During inference, all metrics are computed using sequential computation. The red curve represents results from parallel training, while the blue curve corresponds to sequential training for both training and inference. This comparison confirms the equivalence of parallel and sequential training.

For training equivalence, Figure 10 shows the training and testing curves for a CIFAR classification task with 1024-length inputs using a 5-layer MLP, where each linear layer is followed by a neuron. Inference is performed with sequential computation, while training is conducted using either sequential (blue curve) or parallel (red curve) methods. After 64 training epochs, the training and testing curves align closely, indicating that the gradient computations from both sequential and parallel training are nearly identical. The parallel method achieves an $8.58\times$ speedup with a sequence length of $1,024$.

## L   THE PARALLEL IMPLEMENTATION CASE STATICS DETAILS

The figure presents a comparative analysis of sequential and parallel training processes using the *torch.profiler* tool for a sample case. The trace recording and periods timeline as shown in Figure.11. The corresponding details data statistics in the Table.9 and 10. We designed a simple case using a fully connected layer (FC: $1 \times 10$) and an extremely simple architecture to identify the bottlenecks in sequential and parallel training. We use sequential-MNIST (784 length) with 64 batch size as the input.

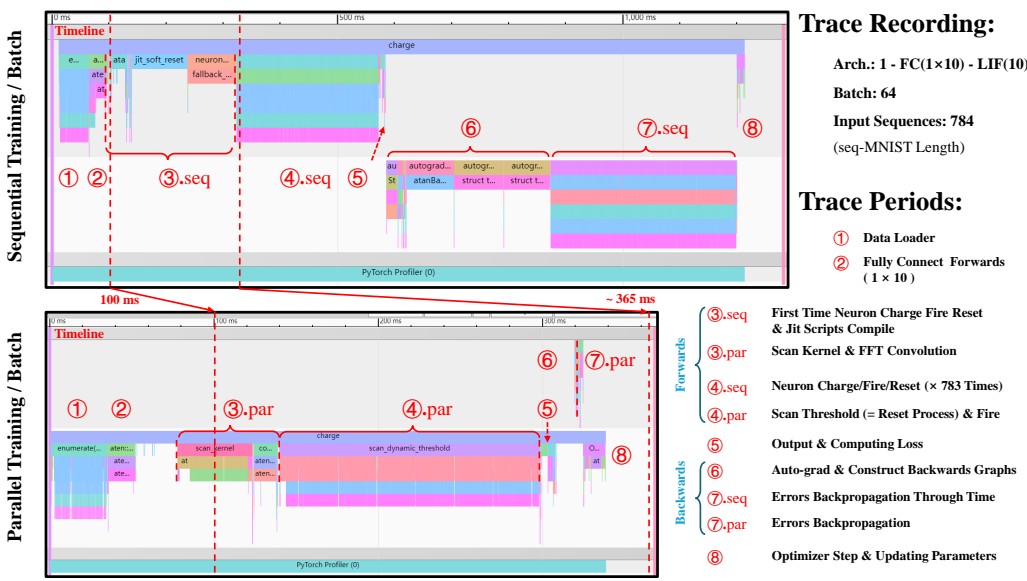

Figure 11: The *torch.profiler* (Paszke et al., 2019) tool was used to visualize runtime during forward and backward propagation. The input length for the sample is 784 sequences with a batch size of 64.

Table 9: Sequential Training Statistics

| Functions | Duration ($\mu s$) | Num. of Calls |
|---|---|---|
| *Forward* | | |
| Charge | 146,725 | 784 |
| Reset | 150,120 | 784 |
| Fire | 91,398 | 784 |
| *Backprop (Autograd)* | | |
| GraphBackward | 250,161 | 1564 |
| AtanBackward | 186,086 | 784 |
| SelectBackward | 91,438 | 784 |
| SubBackward | 20,124 | 787 |
| StackBackward | 19,436 | 1 |
| MmBackward | 1,404 | 1 |
| *Other* | | |
| Other | 244,358 | - |
| **All** | **1,201,250** | **-** |

Table 10: Parallel Training Statistics

| Functions | Duration ($\mu s$) | Num. of Calls |
|---|---|---|
| *Forward* | | |
| Scan Kernel | 45,634 | 1 |
| FFT Conv Op. | 14,627 | 1 |
| Scan Dynamic Thr. | 158,774 | 1 |
| Fire | 459 | 1 |
| *Backprop (Autograd)* | | |
| FftR2CBackward | 1,099 | 1 |
| AtanBackward | 762 | 1 |
| MulBackward | 582 | 1 |
| FftC2RBackward | 235 | 1 |
| MmBackward | 1,615 | 1 |
| *Other* | | |
| Other | 113,863 | - |
| **All** | **337,650** | **-** |

In the sequential training timeline, each phase of forward and backward propagation happens one after the other, resulting in a total training time of approximately $1200ms$. These phases include data loading, fully connected forward passes, neuron charge/fire/reset operations, output computation, autograd construction, and error backpropagation. This sequential computation introduces significant delays, particularly during the repetitive neuron charging and resetting in the backward phases, which dominate the computation time.

In contrast, the parallel training timeline reduces the overall training time to around *337 ms* by executing multiple forward and backward propagation phases simultaneously. This approach leverages parallel processing to handle operations such as neuron charging and threshold scanning concurrently, thereby reducing redundant delays. The comparison highlights significant efficiency gains in processes like autograd, constructing backward graphs, and error backpropagation.

## M  THE STATISTICS OF FIRE RATE

The spiking fire rate is derived from the top-1 test accuracy model (Average in Table 11). We extract the fire rate for each layer (12 to 16). On average, the Spatial Neuron ($\mathcal{SN}$) exhibits a higher fire rate than the PRF ($\mathcal{TN}$). The checkpoints and statistical code can be found in the open-source repository.

Table 11: The average fire rates for each tasks.

| Tasks | ListOps | Text | Retrieval | Image | Pathfinder |
|---|---|---|---|---|---|
| Avg. Fire Rate (%) | 3.53 | 4.32 | 1.48 | 3.47 | 3.29 |

Table 12: The fire rate (%) across different layers on **ListOps** task.

| Layer | 1 | 2 | 3 | 4 | 5 | 6 | 7 | 8 |
|---|---|---|---|---|---|---|---|---|
| $\mathcal{TN}$ | 0.0 | 5.17 | 2.50 | 2.83 | 0.80 | 1.17 | 3.02 | 2.22 |
| $\mathcal{SN}$ | 9.60 | 5.29 | 4.51 | 2.63 | 5.58 | 3.57 | 9.57 | 5.07 |

Table 13: The fire rate (%) across different layers on **Text** task.

| Layer | 1 | 2 | 3 | 4 | 5 | 6 |
|---|---|---|---|---|---|---|
| $\mathcal{TN}$ | 2.11 | 4.30 | 2.79 | 1.66 | 1.02 | 1.48 |
| $\mathcal{SN}$ | 9.20 | 9.90 | 10.11 | 7.18 | 5.55 | 5.25 |

Table 14: The fire rate (%) across different layers on **Retrieval** task.

| Layer | 1 | 2 | 3 | 4 | 5 | 6 |
|---|---|---|---|---|---|---|
| $\mathcal{TN}$ | 0.66 | 0.42 | 0.42 | 0.49 | 0.48 | 0.87 |
| $\mathcal{SN}$ | 4.79 | 4.24 | 1.54 | 2.46 | 2.50 | 1.91 |

Table 15: The fire rate (%) across different layers on **Image** task.

| Layer | 1 | 2 | 3 | 4 | 5 | 6 |
|---|---|---|---|---|---|---|
| $\mathcal{TN}$ | 0.22 | 7.10 | 5.25 | 3.67 | 3.90 | 4.24 |
| $\mathcal{SN}$ | 0.44 | 9.90 | 5.71 | 4.35 | 2.77 | 1.07 |

Table 16: The fire rate (%) across different layers on **Pathfinder** task.

| Layer | 1 | 2 | 3 | 4 | 5 | 6 |
|---|---|---|---|---|---|---|
| $\mathcal{TN}$ | 3.37 | 3.96 | 5.07 | 4.53 | 4.53 | 4.78 |
| $\mathcal{SN}$ | 3.66 | 2.98 | 3.74 | 3.63 | 3.33 | 2.53 |

# N    THE ABLATION EXPERIMENTS

We examine the sensitivity of the $\Delta$ and $\theta$ hyper-parameters during initialization. Using a neural network architecture with layers sized 1-64-256-256-10 and a batch size of 256, we fixed $\tau = 2$ and set $\Delta$ and $\theta$ as non-trainable scale values. Figure 12 illustrates this sensitivity for the sMNIST *(left)* and psMNIST *(right)* datasets. The gray frames highlight a shift in sensitive regions from sMNIST to psMNIST, indicating that different data distributions require careful initialization of $\Delta$ and $\theta$. Furthermore, we investigate the impact of varying initialization of $\theta$ values on the performance of the LRA tasks, as shown in Tables 17 - 21. The suitable initialization of $\theta$ is crucial.

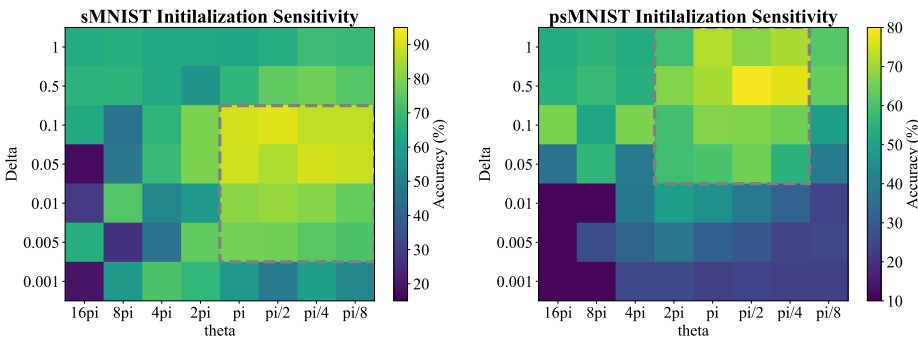

Figure 12: The comparison of sMNIST (***left***) and psMNIST (***right***) under the different initialization of $\theta$ and $\Delta$ hyper-parameters after 50 epoches training (The $\theta$ and $\Delta$ is scale value without training).

Table 17: ListOps (2048) accuracy for different $\theta$ ranges initialization.

| $\theta$ | $[0, \pi/4]$ | $[0, \pi/5]$ | $[0, \pi/6]$ | $[0, \pi/7]$ | $[0, \pi/8]$ |
|---|---|---|---|---|---|
| Acc. (%) | 56.85 | 55.60 | **59.20** | 58.05 | 54.55 |

Table 18: Text (4096) accuracy for different $\theta$ range initialization.

| $\theta$ | $[0, \pi]$ | $[0, \pi/2]$ | $[0, \pi/4]$ | $[0, \pi/8]$ | $[0, \pi/16]$ |
|---|---|---|---|---|---|
| Acc. (%) | 81.27 | 83.28 | 84.94 | **86.33** | 85.32 |

Table 19: Retrieval (4000) accuracy for different $\theta$ range initialization.

| $\theta$ | $[0, \pi/4]$ | $[0, \pi/5]$ | $[0, \pi/6]$ | $[0, \pi/7]$ | $[0, \pi/8]$ |
|---|---|---|---|---|---|
| Acc. (%) | 89.64 | 89.52 | 89.75 | **89.88** | 89.72 |

Table 20: Image (1024) accuracy for different $\theta$ range initialization.

| $\theta$ | $[0, 2\pi/0.15]$ | $[0, 2\pi/0.2]$ | $[0, 2\pi/0.25]$ | $[0, 2\pi/0.3]$ | $[0, 2\pi/0.35]$ |
|---|---|---|---|---|---|
| Acc. (%) | 84.36 | 84.43 | **84.77** | 84.54 | 84.38 |

Table 21: Pathfinder (1024) accuracy for different $\theta$ range initialization.

| $\theta$ | $[0, 2\pi]$ | $[0, 2\pi/0.8]$ | $[0, 2\pi/0.6]$ | $[0, 2\pi/0.4]$ | $[0, 2\pi/0.2]$ |
|---|---|---|---|---|---|
| Acc. (%) | 91.14 | 91.19 | **91.76** | 89.87 | 90.97 |

## O  THE COMPARISON OF THE DIFFERENT NEURON MODELS

This section provides a comprehensive comparison of various neuron models across three key aspects: feature differences, dynamic and energy efficiency, and parallel reset mechanisms. It highlights differences in computational capabilities, efficiency, and implementation methods, offering a clear overview of their strengths and limitations. Finally, we give the insight of the connection between the SSMs and PRF.

### O.1  THE OVERVIEW COMPARISON OF FEATURES

Firstly, we overview the comparison of various neuron models based on their key features, such as support for parallel training, input cache-free operation, oscillation behavior, and element-wise multiplication. Table.22 highlights how these models differ in their operational and computational capabilities.

Table 22: The Comparison of Different Features.

| Neuron Models | Parallel Training | Input Cache Free | Oscillation with $V_{mem}$ | Element-Wise Mul. |
|---|---|---|---|---|
| LIF | No | **Yes** | No | **Yes** |
| Masked PSN (Fang et al., 2024) | **Yes** | No | No | - |
| PSN (Fang et al., 2024) | **Yes** | No | No | - |
| Masked PSN (Fang et al., 2024) | Yes | Partial (When $k = 1$ Yes) | No | - |
| Sliding PSN (Fang et al., 2024) | **Yes** | Partial (When $k = 1$ Yes) | No | - |
| PMSN (Chen et al., 2024) | Partial | **Yes** | Partial | Partial |
| adLIF (Baronig et al., 2024) | No | **Yes** | **Yes** | **Yes** |
| Parallelizable LIF (Yarga & Wood, 2024) | **Yes** | **Yes** | No | **Yes** |
| **PRF (Ours)** | **Yes** | **Yes** | **Yes** | **Yes** |

### O.2  THE COMPARISON OF DYNAMIC AND ENERGY COST

This subsection analyzes the dynamics and theoretical energy costs of different neuron models. Table.23 provides a detailed breakdown of the mathematical formulations for each model's dynamics, alongside their respective theoretical energy costs.

Table 23: Comparison of model dynamics and their theoretical energy costs. The symbols $*$ means the analysis mainly refer from PMSN (Chen et al., 2024).

| Neuron Models | Dynamics | Theoretical Energy Cost |
|---|---|---|
| LIF$^*$ | $V[t] = (1 - \frac{1}{\tau})V[t-1] + I[t] - \theta S[t-1]$ | $hmtFr_{\text{in}}E_{\text{AC}} + mtE_{\text{MAC}}$ |
| PSN$^*$ (Fang et al., 2024) | $V[t] = \sum_{i=0}^{t} W_{t,i}I[i]$ | $hmtFr_{\text{in}}E_{\text{AC}} + mt^2E_{\text{MAC}}$ |
| Masked PSN$^*$ (Fang et al., 2024) | $V[t] = \sum_{i=t-k+1}^{t} W_{t,i}I[i]$ | $hmtFr_{\text{in}}E_{\text{AC}} + kmtE_{\text{MAC}}$ |
| Sliding PSN$^*$ (Fang et al., 2024) | $V[t] = \sum_{i=t-k+1}^{t} W_i I[i]$ | $hmtFr_{\text{in}}E_{\text{AC}} + kmtE_{\text{MAC}}$ |
| PMSN$^*$ (Chen et al., 2024) | $V_h[t] = \bar{\tau}V_h[t-1] + \Phi_c I[t]$ 
 $I_h[t] = \Phi_b V_h(t) + \gamma_n I(t)$ 
 $v_s[t] = v_s[t-1] + I_h[t] - \theta S[t-1]$ | $hmtFr_{\text{in}}E_{\text{AC}} + 8(n-1)mtE_{\text{MAC}}$ |
| adLIF (Baronig et al., 2024) | $\hat{u}[t] = \alpha u[t-1] + (1-\alpha)(-w[t-1] + I[t])$ 
 $w[t] = \beta w[t-1] + (1-\beta)(a\hat{u}[t-1] \cdot (1 - S[t-1]) + bS[t])$ | $hmtFr_{\text{in}}E_{\text{AC}} + 6mtE_{\text{MAC}}$ |
| **PRF (ours)** | $\tilde{u}[t] = \exp\left(\Delta \cdot (-\frac{1}{\tau} + i \cdot \theta)\right)\tilde{u}[t-1] + \Delta I[t]$ | $hmtFr_{\text{in}}E_{\text{AC}} + 5mtE_{\text{Mul}} + 3mtE_{\text{AC}}$ |

$h$ - input dimension, $m$ - neuron numbers, $t$ - simulation time, $k$ - order of PSN families,
$Fr_{\text{in}}$ - average spike frequency of each presynaptic layer, $n$ - compartment number of our PMSN,

### O.3  THE COMPARISON OF PARALLEL RESET METHOD

This subsection examines the parallel methods for resetting mechanisms proposed in different works as shown in Table.24. This offer insights into their time complexity and equivalence.

Table 24: The Comparison of Parallel Methods for Resetting Mechanism.

| Methods | Integrate-Leaky Process | Scan Reset Process | Total Complexity | Equivalent with Sequential |
|---|---|---|---|---|
| PMSN (Chen et al., 2024) | $O(L \cdot \log L)$ | Prefix sum with $O(2 \cdot \log L)$ | $O((L+2) \cdot \log L)$ | Partial (Only when positive input) |
| Parallelizable LIF (Yarga & Wood, 2024) | $O(L \cdot \log L)$ | **Without Reset** | $O(L \cdot \log L)$ | **Yes** |
| **Decoupled Reset (Ours)** | $O(L \cdot \log L)$ | Dynamic Scan with $O(L)$ | $O(L \cdot (\log L + 1))$ | **Yes** |

## O.4 THE CONNECTION BETWEEN SSMs AND PRF.

Both PRF and Structured SSMs (State-Space Models) are subsets of SSMs, sharing the same general abstract formulation:

$$u_t = \bar{A} u_{t-1} + \bar{B} c_t, \quad y_t = f(u_t).$$

The similarity is that both frameworks involve: $A$: State Transition, $B$: Input Transformation and $f(u_t)$: Output Function. While the main differences is the dynamic process and transition dimension. The detail is as shown in Table.25.

Table 25: Comparison Between PRF and Structured SSMs.

| Component | Structured SSMs | PRF |
|---|---|---|
| $\bar{A}$ | $\exp(\Delta A) \in \mathbb{R}^{h \times h}$ | $\exp(\Delta \cdot (-\frac{1}{\tau} + i\theta)) \in \mathbb{C}^d$ |
| $\bar{B}$ | $A^{-1} \left( e^{A\Delta} - I \right) B \in \mathbb{R}^{h \times d}$ | $\Delta \in \mathbb{R}^d$ |
| $f(u_t)$ | $C \cdot u_t$ 
 $C \in \mathbb{R}^{d \times h}, f : \mathbb{R}^h \to \mathbb{R}^d$ | $\mathcal{H}(\Re(u_t) - V_{th})$ 
 $f : \mathbb{R}^d \to \mathbb{R}^d$ |

## P  SIMPLIFIED CODE FOR IMPLEMENTATION

**Listing 1**. Simplified PyTorch-like Implementation of SD-TCM Module.

```python
class SD_TCM(nn.Module):
    def __init__(self, d_model, dropout=0.0, **kernel_args):
        super().__init__()
        self.h = d_model
        # whether trainable amplitude, ref Sec 4.4 and Table 4.
        self.train_amp = kernel_args.get('train_amp', False)
        # whether bidirectional architecture, ref Eq.102
        self.bidirectional = kernel_args.get('bidirectional', False)

        self.neuron1 = PRF(channels=self.h)
        self.dropout1 = dropout_fn(dropout)

        if self.bidirectional:
            self.reverse_neuron1 = PRF(channels=self.h)
            self.pro_linear1 = nn.Linear(2 * self.h, self.h)
        else:
            self.pro_linear1 = nn.Linear(self.h, self.h)

        self.neuron2 = surrogate.ATan()
        self.dropout2 = dropout_fn(dropout)
        self.pro_linear2 = nn.Linear(self.h, self.h)

        if self.train_amp:
            nn.Parameter(torch.log(torch.ones(1)))  # only one parameter
            alpha = torch.log(torch.ones(1))
            self.register("alpha", alpha, 0.001)

    def forward(self, u):
        """ Input and output shape (T, B, D) """
        s = self.neuron1(u)                   # (T B D)
        if self.bidirectional:
            rev_s = self.reverse_neuron1(u.flip(dims=[0])).flip(dims=[0])
            s = torch.concat([s, rev_s], dim=-1)
        y = self.pro_linear1(self.dropout1(s))
        x = y + u

        s = self.neuron2(x - 0.5)
        if self.train_amp:
            s = s * torch.exp(self.alpha))  # {0, 1} * trainable alpha
        y = self.pro_linear2(self.dropout2(s)) + x
        return y

    def register(self, name, tensor, lr=None):
        """Register a tensor with a
        configurable learning rate and 0 weight decay"""
        if lr == 0.0:
            self.register_buffer(name, tensor)
        else:
            self.register_parameter(name, nn.Parameter(tensor))

            optim = {"weight_decay": 0.0}
            if lr is not None: optim["lr"] = lr
            setattr(getattr(self, name), "_optim", optim)
```

**Listing 2**. Simplified PyTorch-like Implementation of PRF Neuron.

```python
class PRF(nn.Module):
    def __init__(self, channels, tau, v_threshold, surrogate_function,
            fr_scale: float=1., dt_min: float=0.1, dt_max: float=0.001):
        super().__init__()
        self.channels = channels         # int with D
        self.tau = tau                    # float defaut=2.0
        self.fire_fn = surrogate_function
        self.threshold = v_threshold      # float defaut=1.0

        u1, u2 = torch.rand(channels), torch.rand(channels)
        max_phase = 2 * torch.pi

        # fr_scale for controling range of unifrom
        log_theta = torch.log(max_phase * u2 / fr_scale)
        log_dt = u1 * (math.log(dt_max) - math.log(dt_min))
                    + math.log(dt_min)

        # often setting no weight decay and idenpendent learning rate
        self.log_dt = nn.Parameters(log_dt)           # (D)
        self.log_theta = nn.Parameters(log_theta)  # (D)

    def sequential_step(x, v, tau, dt, theta):
        """
        x     : (B, D)      Input with B batch, D dimension
        v     : (B, D)      Previous hidden State
        tau   : float       self.tau
        dt    : (D)         torch.exp(self.log_dt)
        theta : (D)         torch.exp(self.log_theta)
        """
        v = torch.exp(dt * (-1 / tau + 1j * theta)) * v + dt * x
        spike = heaviside(v.real - self.v_threshold)
        return spike, v

    def parallel_step(self, x):
        """
        x    : (T, B, D)  Input with T Sequence, B Batch, D Dimension
        s_seq: (T, B, D)  Output
        """
        dt, theta = torch.exp(self.log_dt), torch.exp(self.log_theta)
        beta = torch.exp(dt * (- 1/self.tau + 1j * theta))
        kernel = self.scan_kernel(beta, dt, T)          # (T, D)
        u_seq = self.charge(kernel, x)                   # (T, B, D)
        s_seq = self.surrogate_function(u_seq.real - self.v_threshold)
        return s_seq

    def charge(self, kernel, input_seq):
        T, D = kernel.shape
        kernel_expand = kernel.squeeze().view(T, 1, D).contiguous()
        output_fft = torch.fft.ifft(
            torch.fft.fft(kernel_expand, n=2 * T, dim=0)
            * torch.fft.fft(input_seq, n=2 * T, dim=0), n=2* T, dim=0)
        u_seq = output_fft[:T]
        return u_seq.real

    def scan_kernel(beta, dt, T):
        K = beta.unsqueeze(-1) ** torch.arange(T)   # (D, T)
        B = dt.unsqueeze(-1)                          # (D, 1)
        return (K * B).T
```

