# OpenReview forum: "PRF: Parallel Resonate and Fire Neuron for Long Sequence Learning in Spiking Neural Networks"
_ICLR.cc/2025/Conference — Submitted to ICLR 2025_

### Official Review · Reviewer_vmHd · 2024-10-15

**Soundness:** 2
**Presentation:** 2
**Contribution:** 2
**Rating:** 3
**Confidence:** 4

**Summary:**

This paper proposes a new decoupled reset method to accelerate training, making the effect of parallel training equivalent to sequential training. This method can increase the back propagation speed by three orders of magnitude and is applicable to all types of spiking neurons. In addition, in order to effectively capture long-term dependencies, this paper designs a new type of PRF neuron, which has an oscillating membrane potential in the complex domain and works through an adaptive differentiable reset mechanism.

**Strengths:**

1. This paper integrate PRF neurons into the SD-TCM module, which performs comparable to S4 in long-term dependency tasks while reducing inference energy consumption by two orders of magnitude.
2. Improved the previous SNN parallel method to achieve acceleration, and achieved good results on the lra dataset.

**Weaknesses:**

1. Since there is no code in the appendix of this article, I have doubts about the authenticity of lra's experimental results.
2. There are too many typos in the article. The citation format of the whole article should be “\citep{}” instead of  “\cite{}”. At the same time, please use the same “spike-driven” format throughout the article.
3. The comparison in Figure 1 is unfair, as SSM is a block and the proposed PRF is just a neuron. This makes the complexity comparison unfair.
4. In addition, I have doubts about the effectiveness of the proposed method. In Table 4, we can see that the improvement gains of different modules are very low. Does this mean that the proposed method relies heavily on its own architecture? Please add additional ablation experiments on other architectures.

**Questions:**

see weakness bellow.

---

> ### Author Response · Authors · 2024-11-23
> **Reply  (1/2)**
>
> ----
> Dear Reviewer vmHd,
>
> We sincerely appreciate your valuable feedback and thoughtful suggestions. We also greatly appreciate the time and effort you have dedicated to evaluating our work. In response to your comments, we will address each of the concerns and questions you raised individually and in detail.
>
> ----
>
> > **W1: Since there is no code in the appendix of this article, I have doubts about the authenticity of lra's experimental results.**
>
> **Reply:** We sincerely thank you for your valuable feedback. To improve code reproducibility, we have included a simplified PyTorch-like implementation of PRF and SD-TCM in Appendix P (pages 31–32 of the manuscript). Additionally, we will make the model checkpoint available alongside the open-source repositories.
>
> ----
>
> > **W2: There are too many typos in the article. The citation format of the whole article should be “\citep{}” instead of “\cite{}”. At the same time, please use the same “spike-driven” format throughout the article.**
>
> **Reply:** We sincerely appreciate your careful review and thank you for pointing out the typos and formatting issues. We have ensured consistency in the citation format (\citep{}) throughout the manuscript. We have also corrected the term "spike-driven." All changes are highlighted in blue in the revised manuscript for easy reference.
>
>
> ----
>
> > **W3: The comparison in Figure 1 is unfair, as SSM is a block and the proposed PRF is just a neuron. This makes the complexity comparison unfair.**
>
> **Reply:** Thank you for your insightful comments. We initially intended to highlight that directly adding the spike function to the SSM as the neuron leads to higher complexity compared to using individual neurons and we agree with the reviewer that this explanation might be misleading.
>
> To address your concern, we provide a more comprehensive comparison of the complexities across various blocks, including a detailed breakdown of the complexities of the token mixing layer and channel mixing layer for different modules (S4, Binary S4D, and Ours). As shown in Table R1, the complexity is significantly reduced in Token Mixing. In fact, the Token Mixing complexity is approximately halved, as illustrated in Table R1. We have also added this comparison in Table 7 of Appendix I of the manuscript.
>
> *Table R1. The comparision of inference complexity. (where $R$ denotes the spike firing rates $\in (0, 1)$, $\sigma$: Sigmoid activation function, $\mathcal{H}$: Heaviside function and $\Re$: real part of the complex number, and Ter means the ternary output $\\{-1, 0, 1\\}$ with a dynamic threshold).*
> |  | Token Mixing  |   | Channel Mixing  |   |
> | ---  | ---  | ---  | ---  | ---  |
> |   | Computation  | Complexity  |  Computation  | Complexity  |
> | **S4-LegS** [1]  |  $ \boldsymbol{u} _ t = A \boldsymbol{u}_{t-1} + B\boldsymbol{x}_t $   |  $ O(H^2 + DH) $  | $ \boldsymbol{n}_t = \textbf{Linear}_1(\boldsymbol{y} _ t) $  | $ O(D^2) $  |
> |              |  $ \boldsymbol{y}_t = C \boldsymbol{u}_t + D \boldsymbol{x}_t $ | $ O(HD + D^2) $ |   $ \boldsymbol{m}_t = \textbf{Linear}_2(\boldsymbol{y} _ t) $  | $ O(D^2) $  |
> |              |   |   |  $ \boldsymbol{o}_t = \boldsymbol{n} _ t \odot \sigma(\boldsymbol{m} _ t) $  | $ O(3D) $  |
> | | | | | |
> | **Binary S4D** [2]  |  $ \boldsymbol{u} _ t = A \boldsymbol{u}_{t-1} + B\boldsymbol{x}_t $   |  $ O(H^2 + DH) $  | $ \boldsymbol{n}_t = \textbf{Linear}_1(\boldsymbol{y} _ t) $  | $ O(R \cdot D^2) $  |
> |              |  $ \boldsymbol{y}_t = \mathcal{H}(C \boldsymbol{u}_t + D \boldsymbol{x}_t )$ | $ O(HD + D^2) $ |   $ \boldsymbol{m}_t = \textbf{Linear}_2(\boldsymbol{y} _ t) $  | $ O(R \cdot D^2) $  |
> |              |   |   |  $ \boldsymbol{o}_t = \boldsymbol{n} _ t \odot \sigma(\boldsymbol{m} _ t) $  | $ O(3D) $  |
> | | | | | |
> | **GSU**  [2] |  $ \boldsymbol{u} _ t = A \boldsymbol{u}_{t-1} + B\boldsymbol{x}_t $   |  $ O(H^2 + DH) $  | $ \boldsymbol{n}_t = \textit{Ter}(\boldsymbol{W} _ 1 ) \boldsymbol{y} _ t + \boldsymbol{b} _ 1  $  | $ O(R \cdot D^2) $  |
> |              |  $ \boldsymbol{y}_t = C \boldsymbol{u}_t + D \boldsymbol{x}_t $ | $ O(HD + D^2) $ |   $ \boldsymbol{n}_t = \boldsymbol{W} _ 2  \textit{Ter}( \boldsymbol{y} _ t ) + \boldsymbol{b} _ 2 $  | $ O(R \cdot D^2) $  |
> |              |   |   |  $ \boldsymbol{o}_t = \text{GeLU}(\boldsymbol{n} _ t \odot \boldsymbol{m} _ t ) $  | $ O(3D) $  |
> | | | | | |
> | **SD-TCM (Ours)**  | $ \boldsymbol{u}_ t = A \odot \boldsymbol{u}_{t-1} + B \odot \boldsymbol{x}_t $   | $ O(5D) $ | $ \boldsymbol{s}_ t = \mathcal{H}(\boldsymbol{n}_ t - V_ {th}) $ |  --   |
> |              |  $ \boldsymbol{s}_ t = \mathcal{H}(\Re \\{ \boldsymbol{u}_ t \\} - V_{th})) $ |  -  |   $ \boldsymbol{m}_ t = \textbf{Linear}_ 2(\boldsymbol{y}_ t) $  | $ O(R \cdot D^2) $  |
> |              | $ \boldsymbol{n}_ t = \text{Linear}_ 1(\boldsymbol{s}_ t) + \boldsymbol{u}_ t $ | $ O(R \cdot D^2 + D) $ |   |  |

---

> > ### Author Response · Authors · 2024-11-23
> > **Reply (2/2)**
> >
> > ----
> >
> > > **W4: we can see that the improvement gains of different modules are very low. Does this mean that the proposed method relies heavily on its own architecture? Please add additional ablation experiments on other architectures.**
> >
> > **Reply:** Thank you for your insightful question. We have expanded the ablation study on different architectures, as shown in Table R2. Our experiments show that for text tasks, a unidirectional architecture performs better, which could be attributed to the causal nature of text. In contrast, the bidirectional architecture is more effective for image sequences, as the image sequences lack inherent sequential relationships and do not follow a causal order like text data.
> >
> > *Table R2. Performance of different architectures on text and image tasks.*
> > | **Architecture** \ **Task** | **Text** (T = 4096) | **Image** (T = 1024) |
> > |  ----  | ----  | ---- |
> > | Causal | 86.33 | 81.99 |
> > | Bidirectional | 85.60 | 84.77 |
> >
> > ----
> >
> > Reference:
> >
> > [1] Gu A, Goel K, Re C. Efficiently Modeling Long Sequences with Structured State Spaces. ICLR, 2022.
> >
> > [2]  Stan M I, Rhodes O. Learning long sequences in spiking neural networks. Scientific Reports, 2024, 14(1): 21957.
> >
> >
> > ----
> >
> > We sincerely appreciate and thank you again for your valuable review and feedback. If any concerns or questions arise during the discussion period, we would be happy to provide further clarification. We would also be grateful if you could consider a more positive evaluation of our work once all concerns have been addressed.
> >
> > -- Authors

---

> ### Comment · Reviewer_vmHd · 2024-11-25
>
> I appreciate your response. However, there might be some misunderstandings that need to be clarified. Firstly, it's essential to distinguish clearly between a single layer and a neuron. Following that, while bidirectional attention mechanisms are evidently more effective in images than in text, my inquiry pertains to the impact of different architectures like backbone not Causal or the Bidirectional on the PRF.

---

> > ### Comment · Reviewer_vmHd · 2024-11-28
> >
> > From the Ablation 1-3 experiments, it is clear that the chosen model architecture has positively contributed to performance improvement. However, the authors have not provided a detailed experimental analysis of the role of neurons in the model. In this regard, I believe it would be beneficial to include an additional ablation experiment replacing LIF neurons to further assess the impact of different neuron models on overall performance. If the authors are unable to address this concern, I may consider lowering my score.

---

> > > ### Author Response · Authors · 2024-11-29
> > > **Discussion Reply (2/2)**
> > >
> > > Dear Reviewer vmHd,
> > >
> > > Thank you for your valuable suggestions. We agree that we should include additional ablation studies to strengthen our analysis. We apologize for not addressing your previous comment more promptly, as we hope to comprehensively supplement and expand our experiments to better address your concerns.
> > >
> > > Based on SD-TCM architecture, we evaluate the performance of the PRF compared to SPSN and LIF neurons by interchanging them in the token mixing layer. The results after interchanging PRF with LIF and SPSN are shown in Table D1 (Experiments III and V).
> > >
> > > The design of the SD-TCM token mixing layer, which incorporates the hidden state (i.e., the membrane potential dynamics of neurons), is aimed at extracting in-context relationships. In contrast, the channel mixing layer, which does not use a hidden state (employing a Heaviside function (SN) as the neuron), is designed to output useful information, to amplify the long-range learning ability of token mixing.
> > >
> > > With channel mixing, the PRF enhances long-term learning and helps extract more useful information than both LIF and SPSN, as shown in the results for SD-TCM, Experiment I, and Experiment III in Table D4. The necessity of token mixing is further highlighted in Ablations 2 and 3. Without token mixing, the model fails to function effectively.
> > >
> > >
> > > *Table D1. The neuron and module ablation studies of the SD-TCM module.*
> > > | Task: ListOPs (T=2048) | Token Mixing	| Channel Mixing | Acc (%) | Param. (k) | Avg. Train. Time / Epoch |
> > > | --- | --- | --- | --- | --- | --- |
> > > | SD-TCM  | PRF + Linear | SN + Linear  | 59.20 | 271 | 4 min 10 s |
> > > |Ablation 1 | PRF + Linear | Identity | 46.50 |139 | 3 min 45 s |
> > > | Exp. I | SPSN (k=8) [1]+ Linear | SN + Linear | 45.90 | 269 | 24 min 44 s |
> > > | Exp. II | SPSN (k=8) [1]+ Linear | Identity | 45.00 | 137 | 23 min 43 s |
> > > | Exp. III | LIF + Linear | SN + Linear | 44.70 | 269 | 23 min 47 s |
> > > | Exp. V | LIF+ Linear | Identity | 38.50 | 137  | 21 min 35 s |
> > > | Ablation 2 | Identity | SN + Linear | 20.05 | 137 | 1 min 28 s |
> > > | Ablation 3 | Identity | Identity | 20.45 | 5.6 | 1 min 08 s |
> > >
> > > Moreover, to demonstrate the effectiveness and applicability of the PRF neuron beyond the TD-SCM architecture, we conducted an evaluation using both MLP and SSM architectures, as shown in Tables D2 and D3. Specifically, we evaluated the PRF neuron using an MLP architecture, as shown in Table D2, which further supports its effectiveness in basic architectures. We also extended our experiments by incorporating the PRF neuron into a more sophisticated SSM architecture, as presented in Table D3. In SSM architectures, the PRF neuron outperforms other spike functions (LIF [2], Heaviside [3], and Stochastic Firing Function [4]). However, we observed that directly applying the PRF neuron to the SSM architecture led to significant overfitting: after 30 epochs, we achieved a training accuracy of 90.23%, but the test accuracy remained capped at 83.25%. Although we have not yet resolved this overfitting issue due to time constraints, we hypothesize that the PRF neuron, with its extended capacity, may introduce redundancy into the SSM architecture. This potential redundancy requires further investigation and optimization.
> > >
> > >
> > > *Table D2. The experiments by incorporating PRF neuron on MLP and SSMs architecture..*
> > > | MLP + Neuron | psMNIST (T=784) |
> > > | --- | --- |
> > > | **MLP + LIF** | 80.26 % |
> > > | **MLP + PSN**  | 97.76 % |
> > > | **MLP + PMSN** | 97.78 % |
> > > | **MLP + PRF** | **97.90 %** |
> > >
> > >
> > > *Table D3. The LRA on Text (T=4096) experiment with various neuron models with SSM architecture.*
> > > | SSMs + Neuron | LRA on Text (T=4096) |
> > > | --- | --- |
> > > | **SSMs + LIF** [2] |80.41 % |
> > > | **SSMs + Heaviside** [3] | 82.50 % |
> > > | **SSMs + Stochastic Firing Function** [4] | 77.62 % |
> > > | **SSMs + PRF** | **83.25 %** |
> > >
> > > ---
> > >
> > > [1] Fang, et al. Parallel Spiking Neurons with High Efficiency and Ability to Learn Long-term Dependencies. NeurIPS, 2024.
> > >
> > > [2] Shen S, Wang C, Huang R, et al. Spikingssms: Learning long sequences with sparse and parallel spiking state space models[J]. arXiv preprint arXiv: 2408.14909, 2024.
> > >
> > > [3] Stan M I, Rhodes O. Learning long sequences in spiking neural networks. Scientific Reports, 2024, 14(1): 21957.
> > >
> > > [4] Bal, et al. P-SpikeSSM: Harnessing Probabilistic Spiking State Space Models for Long-Range Dependency Tasks. arXiv preprint arXiv: 2406.02923, 2024.
> > >
> > > ---
> > >
> > > We sincerely hope that these responses can help address your concerns. Once again, we greatly appreciate your valuable comments and suggestions, which have significantly contributed to improving the quality of our manuscript. Thank you again for your valuable feedback and review.

---

> > > ### Author Response · Authors · 2024-12-02
> > >
> > > Dear reviewer vmHd,
> > >
> > > We sincerely thank you for your thoughtful and detailed review of our manuscript. Your feedback has been invaluable in guiding our revisions.
> > >
> > > To comprehensively address your concerns, we have conducted additional experiments and made significant revisions to the manuscript, as outlined in our prior response on Nov 30th. Specifically, we have:
> > >
> > > - Clarified the distinction between single-layer comparisons and neuron-level comparisons (revised Table 1 and Table 6).
> > >
> > > - Conducted an ablation experiment replacing LIF neurons to evaluate the role of neuron models on overall performance (see Tables D1, D2, and D3).
> > >
> > > - Extended our analysis of PRF neurons across different architectures, including MLP and SSMs, to further assess their impact and effectiveness.
> > >
> > > We sincerely apologize for the delay in providing these updates, which may have influenced your decision to lower the score. This delay was due to the time required to implement and thoroughly evaluate the SSMs with LIF, as well as the ablation experiments replacing LIF. This process involved significant effort in both implementation and training. The SSMs+PRF implementation and experiments alone took nearly 2 days. For Experiments II to V in Table D1, the training took approximately 3 days, as each epoch took around 23 minutes, and each task required training for 40 epochs.
> > >
> > > We understand that your score was lowered from 5 to 3 shortly before we submitted our previous response on Nov 30th. We sincerely hope that, after reviewing the additional experiments and revisions we have provided, you might reconsider your evaluation. We greatly appreciate your time and would be happy to address any remaining concerns to ensure that all issues are thoroughly resolved. Thank you again for your valuable feedback and consideration.

---

> > ### Author Response · Authors · 2024-11-29
> > **Discussion Reply (1/2)**
> >
> > Dear Reviewer vmHd,
> >
> > Thank you for your valuable feedback. We apologize for the misunderstanding and for not fully addressing your question. We have made the following clarifications:
> >
> > 1. **Firstly**, to ensure a fair comparison and avoid any misunderstanding, we have revised Table 1 from the manuscript and Table 6 from Appendix I to specifically compare the token mixing layer to avoid confusing the layer with the neuron. The revised parts are highlighted in blue.
> >
> > 2. **Secondly**, regarding your comment,``In Table 4, we can see that the improvement gains of different modules are very low``, to avoid any further confusion, we would like to clarify the architecture and experiments presented in Table 4 of our manuscript. The ablation study in Table 4 focuses on the effect of the parameter $\alpha$ (**Only single parameter for each layer**), rather than ablating the layers themselves, therefore the improvement is not significant but it works because of around 0.6% gains from only one parameter. The specific PyTorch-like implementation of $\alpha$ can be found in line 1645 of our revised manuscript.
> >
> >      In SD-TCM, we incorporate PRF into Linear layer to enable long-term learning in the token mixing layer. We use Heaviside function as spatial neuron (SN) in channel mixing as the channel mixing requires no hidden state to extract useful information. We also validate the effectiveness of the token mixing and channel mixing layers by changing with Identity layer (as done in Li et al [1]). A more comprehensive ablation of the modules is summarized in Table D1 and will be included in the revised manuscript.
> >
> > *Table D1. The ablation study of the SD-TCM module.*
> > | | Token Mixing	| Channel Mixing	| Acc (%)	| Param. (k)	| Avg. Train. Time / Epoch|
> > | --- | --- | --- | --- | ---  | --- |
> > |SD-TCM	| PRF + Linear	|  SN + Linear	| 59.20	| 271	| 4 min 10 s |
> > |Ablation1  | PRF + Linear | Identity | 46.50 | 139 | 3 min 45 s |
> > |Ablation2 | Identity	| SN + Linear | 20.05  | 137 | 1 min 28 s |
> > |Ablation3 | Identity | Identity | 20.45 | 5.6 | 1 min 08 s |
> >
> >
> > 3. **Thirdly**, in response to your comment,```The inquiry pertains to the impact of different architectures like backbone, not Causal or the Bidirectional, on the PRF```. we agree that it is important to evaluate the effectiveness of the PRF in different architectures. We have conducted an evaluation using the MLP architecture as shown below in Table D2, demonstrating the efficacy of PRF in basic architecture. Additionally, we extend the experiments by incorporating PRF neuron into SSMs architecture, as shown in Table D3.
> >
> >     In both MLP and SSMs, PRF shows superior performance compared to other neuron models. In Table 3, we compared our results with other SSM variants incorporating spike functions (LIF [2], Heaviside [3], and Stochastic Firing Function [4]) in the SSM architecture [5], and PRF performs better than those neurons. Unfortunately, directly applying PRF to SSMs led to significant overfitting: the experiment shows a training accuracy of 90.23% after 30 epochs while the test accuracy is capped at 83.25%. Due to time constraints, we have not yet resolved the overfitting issue. We tentatively attribute this to the extended capacity of the PRF neuron, which may introduce redundancy into the SSM architecture. Such an issue requires further exploration and optimization.
> >
> >
> > *Table D2. The experiments by incorporating PRF neuron on MLP and SSMs architecture..*
> > | MLP + Neuron | psMNIST (T=784) |
> > | --- | --- |
> > | **MLP + LIF** | 80.26 % |
> > | **MLP + PSN**  | 97.76 % |
> > | **MLP + PMSN** | 97.78 % |
> > | **MLP + PRF** | **97.90 %** |
> >
> >
> > *Table D3. The LRA on Text (T=4096) experiment with various neuron models with SSM architecture.*
> > | SSMs + Neuron | LRA on Text (T=4096) |
> > | --- | --- |
> > | **SSMs + LIF** [2] |80.41 % |
> > | **SSMs + Heaviside** [3] | 82.50 % |
> > | **SSMs + Stochastic Firing Function** [4] | 77.62 % |
> > | **SSMs + PRF** | **83.25 %** |
> >
> >
> > ---
> >
> > [1] Liu, et al. LMUformer: Low complexity yet powerful spiking model with Legendre memory units. ICLR, 2024.
> >
> > [2] Shen S, Wang C, Huang R, et al. Spikingssms: Learning long sequences with sparse and parallel spiking state space models[J]. arXiv preprint arXiv: 2408.14909, 2024.
> >
> > [3] Stan M I, Rhodes O. Learning long sequences in spiking neural networks. Scientific Reports, 2024, 14(1): 21957.
> >
> > [4] Bal, et al. P-SpikeSSM: Harnessing Probabilistic Spiking State Space Models for Long-Range Dependency Tasks. arXiv preprint arXiv: 2406.02923, 2024.
> >
> > [5] Gu A, Goel K, Re C. Efficiently Modeling Long Sequences with Structured State Spaces. ICLR, 2022.
> >
> > ---
> >
> > Thank you for your valuable feedback, which made us realize that some parts of the paper were not clearly explained. Your feedback has been very helpful in improving the quality of our manuscript.

---

### Official Review · Reviewer_QNtT · 2024-10-24

**Soundness:** 3
**Presentation:** 3
**Contribution:** 2
**Rating:** 5
**Confidence:** 4

**Summary:**

This paper proposes a decoupled reset method for parallel training of spiking neurons, significantly improving the training speed of LIF neurons. Additionally, a parallel mechanism is introduced into the Resonate-and-Fire model, combined with the SD-TCM structure to achieve efficient long-sequence learning. This approach significantly reduces model energy consumption while achieving comparable performance to SSM (S4) and outperforming Transformer in LRA tasks. However, the experimental section is incomplete, particularly the ablation studies. Furthermore, the motivation behind the reset mechanism is not clearly described. Additional experiments are recommended to further validate the proposed approach.

**Strengths:**

1. This paper proposes a decoupled reset-based parallel training strategy, reducing the training time complexity of LIF neurons to O(LlogL). The authors claim this is the first application of the reset mechanism in parallel training.

2. The paper introduces a parallel mechanism into Resonate-and-Fire neurons, enhancing the model's temporal processing capability.

3. The paper presents the SD-TCM architecture, which, combined with PRF neurons, was tested on LRA tasks. The authors claim it achieved performance comparable to SSM (S4).

**Weaknesses:**

1. The proposed method's long-sequence learning capability is mainly validated in Section 5.2. However, this validation is limited to the S/PS-MNIST image sequence task, with only marginal improvements compared to current SOTA models.
2. The study lacks effective ablation experiments. In Section 5.3, the model's performance on LRA tasks is evaluated, but the contribution of the SD-TCM structure to the performance was not verified, and the improvement was solely attributed to the temporal processing capability of RF neurons.
3. The paper combines Figures 1(d) and (e) to illustrate the strong temporal processing capability of RF neurons. However, the experimental section lacks relevant visualizations to further explain why RF neurons are more suitable for temporal tasks compared to LIF neurons.
4. The research motivation is unclear. The abstract claims that this is the first implementation of a parallel reset mechanism, yet the experimental section does not evaluate its impact on performance.
5. The reproducibility of the experiments is limited. The experimental section should include multiple trials to further validate the proposed method's effectiveness.

**Questions:**

1. To my knowledge, removing the reset mechanism might be the simplest and most effective approach. Several existing studies have explored parallel implementations, such as PSN[1]. The authors should include an experiment to compare their method with these approaches.
2. The paper validates the proposed method on the S/PS-MNIST dataset and claims to achieve SOTA results. However, the performance on other image sequence datasets is not significant. I suggest that the authors expand the experiments further and compare the proposed method with other advanced sequence models to better demonstrate its advantages.

[1] Fang W, Yu Z, Zhou Z, et al. Parallel spiking neurons with high efficiency and ability to learn long-term dependencies[J]. Advances in Neural Information Processing Systems, 2024, 36.

---

> ### Author Response · Authors · 2024-11-24
> **Reply (1/3)**
>
> Dear reviewer QNtT,
>
> We sincerely appreciate your valuable feedback and thoughtful suggestions. We greatly appreciate the time and effort you have dedicated to evaluating our work. To address your concerns, we will respond to each of the weaknesses and questions you raised individually and in detail.
>
> ---
>
> > **W1: The proposed method's long-sequence learning capability is mainly validated in Section 5.2. However, this validation is limited to the S/PS-MNIST image sequence task, with only marginal improvements compared to current SOTA models.**
>
> > **Q2: The paper validates the proposed method on the S/PS-MNIST dataset and claims to achieve SOTA results. However, the performance on other image sequence datasets is not significant. I suggest that the authors expand the experiments further and compare the proposed method with other advanced sequence models to better demonstrate its advantages.**
>
> **Reply:** Thank you for valuable suggestion and feedback. We have realized that we should have evaluated our neuron model more comprehensively and compared it to other neuron models. In response, we conducted additional experiments on (p)sMNIST, seqCIFAR, and ListOps, as shown in Tables R1, R2, and R3. Within the same architecture, PRF shows the highest accuracy compared to other spiking neuron models on psMNIST (T=784), seqCIFAR (T=1024), and ListOps (T=2048).
>
> *Table R1. The (p)seqMNIST (**T=784**) task, all with 4 layer MLP. The results of PMSN and PSN family mainly refer from Cheng X, et al [1]*
> | Source | New Exp. I | New Exp. II | Table.2 | Cheng X, et al [1] | Cheng X, et al | Cheng X, et al | Cheng X, et al | Table.2 |
> |--- |---|---|---|---|---|---|---|---|
> | Neuron          | PRF      | PRF      | PRF     | PMSN [1]   | PSN [2]    | Masked PSN [2]    | Sliding PSN [2]          | LIF     |
> | Architecture            | With residual and Batch Norm (200 channel) | With residual and Batch Norm (128 channel) | w/o residual and BN (128 channel) | With residual and Batch Norm (208 channel) | With residual and Batch Norm (208 channel) | With residual and Batch Norm (208 channel) | With residual and Batch Norm (208 channel) | w/o residual and BN |
> | sMNIST   | 99.39 %  | 99.29 %  | 99.18 % | **99.53 %**    | 97.90 %   | 97.76 %     | 97.20 %    | 89.28 % |
> | psMNIST      | **97.9 %**   | 97.46 %  | 96.67 % | 97.78 %       | 97.76 %   | 97.53 %  | 82.84 %    | 80.36 % |
> | FLOPs   | 246.73M  | 102.9 M  | 102.2 M | 365.8 M        | 712.7 M    | 262.9 M     | 262.9 M     | 244.3 M |
> | Param.     | 167.0k   | 70.4 K   | 69.9 k  | 156.4 k     | 2.5 M      | 153.7 k       | 52.4 k       | 155.1 k |
>
> *Table R2. The seqCifar (**T=1024**) task. The results of PMSN, PSN and LIF mainly from Cheng X, et al [1]. Since no public code in [1], the specific augmentation techniques used are unclear. On the other hand, [2] employed Mixup and CutMix for seqCIFAR as data augmentation strategies. In contrast, our approach only uses Flip and Cutout as the appropriate data augmentation techniques.*
> | Source     | New Exp. III | New Exp. IV | Cheng X, et al [1] | Cheng X, et al | Cheng X, et al | Cheng X, et al | Cheng X, et al |
> |---|---|---|---|---|---|---|---|
> | Neuron       | PRF         | PRF    | PMSN [1]   | PSN [2]        | Masked PSN [2] | Sliding PSN [2] | LIF    |
> | Arch.    | 6 layer residual block (128 channel) | 6 layer residual block (256 channel) | 6 layer residual block | 6 layer residual block | 6 layer residual block | 6 layer residual block | 6 layer residual block |
> | Acc. (%)   | 82.37%    | **85.33%**     | 82.14%  | 55.24%   | 57.83%   | 70.23%    | 45.07%   |
> |Flops | 584.5 M | 822.65 M | 533.5 M |1640.5 M  | 524.5 M | 524.5 M	| 489.625 M |
> | Param.     | 0.29 M       | 1.1 M      | 0.21 M   | 6.47 M    | 0.38 M   | 0.18 M         | 0.18 M   |
>
> *Table R3. The ListOps (**T=2048**) task. We keep the same training configuration and all hyperparameters, only replacing the neuron. The PSN and Masked PSN [2] do not work for ListOps because these neurons require a fixed T for initialization, whereas the dataset has an unfixed sequence length.*
> | Source       | Table 3 | New Exp. V       |  New Exp. VI          | New Exp. VII |
> |---|----|---|---|---|
> | Neuron  |  PRF  | PSN  [2]   | Masked PSN (k=8) [2]     | Sliding PSN (k=8) [2]  |        |
> | Acc (%)        | **59.20%**       | N.A         | N.A         | 45.90%      |
> | Param. (k)     | 271   | 32.2*1e3           | 32.2*1e3         | 269          |
> | Avg. Train. Time / Epoch |  4 min 10 s        | N.A |  N.A |  24 min 44 s         |

---

> > ### Author Response · Authors · 2024-11-24
> > **Reply (2/3)**
> >
> > > **W2: The study lacks effective ablation experiments. In Section 5.3, the model's performance on LRA tasks is evaluated, but the contribution of the SD-TCM structure to the performance was not verified, and the improvement was solely attributed to the temporal processing capability of RF neurons.**
> >
> > **Reply:** We should have discussed the SD-TCM structure in more detail. To address this, we have added an ablation study on the SD-TCM structure, detailed in Ablations 1-3 in Table R4, to highlight the significance of combining token mixing and channel mixing within SD-TCM.
> >
> > ---
> >
> > > **Q1. To my knowledge, removing the reset mechanism might be the simplest and most effective approach. Several existing studies have explored parallel implementations, such as PSN[1]. The authors should include an experiment to compare their method with these approaches.**
> >
> > **Reply:** Thank you for your valuable suggestion. To determine the influence of the SD-TCM structure and the PRF neuron, we conducted ablation experiments by evaluating various neuron models on the ListOps (T=2048) task, as shown in Ablations 4-6 in Table R3. For a fair comparison, we used the same hyperparameter settings as before, only replacing the mixer or neuron, as done in (Li, et al, 2024 [3]). PRF shows the highest accuracy and training efficiency. Compared to MPSN (k=8), PRF shows higher accuracy and much faster training speed.
> >
> > Note that since PMSN does not provide open-source code, we performed experiments on the PSN series models instead. The PSN series models are excellent work, which are highly effective for image sequence tasks. But for text tasks like ListOps, where the length of T is not fixed, PSN and MPSN cannot be directly applied because these models require a fixed T for initialization. Only SPSN (k-order) (Ablation 6) can be directly applied, as it computes at each time step based on the previous k inputs, requiring a cache for k inputs. While increasing k can improve performance, it also will increases computational load and input caching during inference.
> >
> > *Table R4. The ablation experiments. Keep the same training hyperparameters, only replace block or neuron.*
> > |     | Token Mixing   | Channel Mixing | Acc (%) | Param. (k) | Avg. Train. Time / Epoch                           |
> > |-----------|-------------------------------|----------------|---------|------------|---------------------------------------------------|
> > | SD-TCM    | PRF + Linear                  | SN + Linear    | 59.20   | 271        | 4 min 10 s                                        |
> > | Ablation1 | PRF + Linear                  | Identity       | 46.50   | 139        | 3 min 45 s                                        |
> > | Ablation2 | Identity                      | SN + Linear    | 20.05   | 137        | 1 min 28 s                                        |
> > | Ablation3 | Identity                      | Identity       | 20.45   | 5.6        | 1 min 08 s                                        |
> > | Ablation4 | PSN [1] + Linear              | SN + Linear    | N.A     | 32.2*1e3   | No work, because the neuron requires a fixed T, but the dataset consists of sequences with varying lengths. |
> > | Ablation5 | MPSN(k=8) [1] + Linear        | SN + Linear    | N.A     | 32.2*1e3   | Same with PSN                                     |
> > | Ablation6 | SPSN (k=8) [1] + Linear       | SN + Linear    | 45.90   | 269        | 24 min 44 s                                       |
> >
> > ---
> >
> > > **W3: The paper combines Figures 1(d) and (e) to illustrate the strong temporal processing capability of RF neurons. However, the experimental section lacks relevant visualizations to further explain why RF neurons are more suitable for temporal tasks compared to LIF neurons.**
> >
> > **Reply:** We appreciate your valuable feedback. We should have articulated more clearly the connection between our experiments and Figure 1. The explanations for why RF neurons outperform LIF neurons are provided in the Figure 4 (right) and in Figure 5. Specifically, LIF neurons can be seen as a special case without resonance. When the decay factor is very small ("fast decay"), the model suffers from gradient vanishing issues, as illustrated in the landscape of Figure 5a in the manuscript. This corresponds to the "fast decay" scenario shown in Figure 1c. Conversely, as the decay increases ("slow decay"), although some dependencies are captured, the model faces gradient blurring issues, as shown in the landscape of Figure 5c, which corresponds to Figure 1d. In contrast, RF neurons enable smoother gradients, as illustrated in the landscape of Figure 5d, which aligns with Figure 1e. This also explains why RF neurons are better suited for long-term temporal tasks compared to LIF neurons.

---

> ### Author Response · Authors · 2024-11-24
> **Reply (3/3)**
>
> > **W4: The research motivation is unclear. (1/2)**
>
> **Reply:** We sincerely thank you for your valuable feedback. We will modify the abstract and introduction to make them clearer. Long-sequence modeling is crucial for the machine learning community because it enables the processing and understanding of increasingly complex temporal data, which is essential for tasks such as language modeling, speech recognition, and time-series forecasting. Our work is motivated by two challenges in long-sequence modeling with spiking neural networks: inefficient training due to quadratic complexity in traditional methods (e.g., BPTT) and the inability of standard spiking neurons (e.g., LIF) to capture long-range dependencies. Although some existing parallel neuronal efforts have been applied, they are limited to fixed-length image sequences (typically no longer than 1024 timesteps), and do not examine longer sequences (such as T=2048) or variable-length text datasets. As a result, they cannot extend efficiently to other fields like language modeling. Therefore, we rethink these challenges in the SNN field.
>
> To address these issues, we propose the decoupled reset method for parallel training, which significantly speeds up training, and the Parallel Resonate and Fire (PRF) neuron, which efficiently captures long-range dependencies. Combined, these innovations enable energy-efficient SNNs with performance comparable to state-of-the-art models like S4, reducing inference energy by two orders of magnitude while maintaining high performance. We believe this work advances both the scalability and efficiency of SNNs for long-sequence tasks.
>
>
> ---
>
> > **W4: … The abstract claims that this is the first implementation of a parallel reset mechanism, yet the experimental section does not evaluate its impact on performance.(2/2)**
>
> **Reply:** We sincerely thank you for your valuable feedback. We should have clarified the impact of the reset mechanism on performance more clearly.
>
> For LIF neurons, handling the reset enables parallelization, which accelerates computation without affecting accuracy. This is due to the equivalence between sequential and parallel implementations. The specific performance comparisons are provided in Figures 9 and 10 in Appendix K of the manuscript. We will clarify this comparison further in the revised manuscript.
>
> For the PRF neuron, we eliminate the explicit reset mechanism by implementing oscillation, which can be regarded as an implicit reset mechanism. We conducted additional experiments, as shown in Table R5, and the results indicate that introducing an explicit reset mechanism decreases accuracy. The observed performance degradation suggests that the additional explicit reset limits the neurons' ability to oscillate effectively.
>
>
> *Table R5. The experiments of additional reset applied on PRF.*
> |       | **ListOps (T=2048) Task Acc.**            | **Training times / epoch**                        |
> |-------|--------------------------------------------|---------------------------------------------------|
> | **PRF**   | 59.2 %                                     | 4 min 10 s                                         |
> | **PRF + reset** | 37.2 %                                 | 40 min 28 s
>
> ---
>
> > **W5: The reproducibility of the experiments is limited. The experimental section should include multiple trials to further validate the proposed method's effectiveness.**
>
> **Reply:** Thank you for your valuable suggestion. All of our experiments use the same fixed random seed, ensuring the reproducibility of our results. We also conducted additional multiple trials with different random seeds for sMNIST, as shown in Table R6. We will incorporate these results in the revised manuscript.
>
> *Table R6. The mulitple trials on sMNIST.*
> | Fixed Random Seed | sMNIST Acc. |
> |-------------------|------------|
> | 2222 (Before fix used) | 99.39 %  |
> | 0                 | 99.28 %    |
> | 1                 | 99.35 %    |
> | 2                 | 99.32 %    |
> | 3                 | 99.40 %    |
>
> ---
>
> Reference:
>
> [1] Chen X, Wu J, Ma C, et al. PMSN: A Parallel Multi-compartment Spiking Neuron for Multi-scale Temporal Processing. arXiv preprint arXiv:2408.14917, 2024.
>
> [2] Fang W, Yu Z, Zhou Z, et al. Parallel spiking neurons with high efficiency and ability to learn long-term dependencies. NeurIPS, 2024, 36.
>
> [3] Liu Z, Datta G, Li A, et al. LMUFormer: Low Complexity Yet Powerful Spiking Model With Legendre Memory Units. ICLR, 2024.
>
> ---
>
> We sincerely appreciate and thank you again for your valuable review and feedback. If any concerns or questions arise during the discussion period, we would be happy to provide further clarification. We would also be grateful if you will consider a more positive evaluation of our work once all concerns have been addressed.
>
> -- Authors

---

> > ### Comment · Reviewer_QNtT · 2024-11-28
> >
> > The additional experiments have addressed most of my issues, so I have increased my score.

---

### Official Review · Reviewer_ZwjR · 2024-10-29

**Soundness:** 3
**Presentation:** 4
**Contribution:** 3
**Rating:** 8
**Confidence:** 4

**Summary:**

This paper focuses on SNNs for long sequence learning. It first reformulates the reset mechanism of the LIF model into an iterative form to enable parallel training. The authors then propose the PRF neuron to enhance the modeling of long-range dependencies. Both theoretical and experimental results demonstrate the effectiveness and efficiency of the proposed methods.

**Strengths:**

- The paper is well-written and easy to read, with excellent presentation. The proposed methods are novel.

- The experiments and analyses are comprehensive. The performance improvements on the challenging LRA tasks in terms of both efficiency and accuracy are significant.

- The PRF neuron shows promise for deployment on neuromorphic chips for inference.

**Weaknesses:**

- The reset mechanism cannot be computed in parallel with the membrane potential integration process, indicating that the overall neuron model is not fully trained in parallel.

- There is a lack of comparison with existing parallel spiking neuron models, such as the PSN family models [1] and PMSN [2] models.


-----
After the discussion, all my concerns have been addressed, and I have raised my score to 8.

**Questions:**

- What are the connections between the proposed PRF neuron and SSMs?

- What impact does the reset have on the performance of the PRF model in LRA tasks?

- The existing PSN and PMSN models also aim to address the challenges of parallel training and modeling long-range dependencies. Please discuss the differences between PRF and these models.

[1] Parallel Spiking Neurons with High Efficiency and Ability to Learn Long-term Dependencies.

[2] PMSN: A Parallel Multi-compartment Spiking Neuron for Multi-scale Temporal Processing.

---

> ### Author Response · Authors · 2024-11-24
> **Reply (1/3)**
>
> Dear reviewer ZwjR,
>
> We sincerely appreciate your valuable feedback and thoughtful suggestions. We greatly appreciate the time and effort you have dedicated to evaluating our work. To address your concerns, we will respond to each of the weaknesses and questions you raised individually and in detail.
>
> ---
>
> > **W1: The reset mechanism cannot be computed in parallel with the membrane potential integration process, indicating that the overall neuron model is not fully trained in parallel.**
>
> **Reply:** Thank you for your insightful question. We agree with the reviewer that the reset mechanism is solved sequentially. Nevertheless, our approach enables the input and output operations to proceed in parallel. This avoids the need for the membrane potential update to wait for the previous spike outputs, thereby enhancing computational efficiency. We have listed the time for each block with LIF and Parallelized LIF in the seqMNIST task, as shown in Table R1.
>
> The forward pass benefits from a significant reduction in charge and fire time, with parallelization reducing the total forward runtime by ~43%. This is because the dynamic equations eliminate the need to wait for the previous timestep's spike output during the membrane potential update. The sequential process of dynamic scanning can be accelerated further using techniques such as prefix sum scans or CUDA. We plan to explore these approaches in our ongoing research.
>
> Furthermore, the backward pass shows significant speedups, with the total backward time reduced from 567,245 µs to just 2,678 µs. These improvements are due to parallelizing the LIF input and output, which reduces the complexity of recursive graph construction during the backward pass (auto-grad) compared to sequential LIF. Therefore, on the premise of ensuring equivalence, this method could accelerate both the forward and backward passes in parallel. We hope this clarifies the benefits of our parallelization approach.
>
> *Table R1. The runtime of LIF and Parallelized LIF with 784 sequence length inputs.*
> | Run Time ($\mu s$)                | LIF      | Parallelized LIF |
> |------------------------------|----------|------------------|
> | Forward: Charge               | 146,725  | 60,261           |
> | Forward: Reset                | 150,120  | 158,774          |
> | Forward: Fire                 | 91,398   | 459              |
> | **Forward Total**             | **388,240** | **219,494**     |
> | Backward: Fire (Surrogate)    | 186,086  | 762              |
> | Backward: Membrane Part       | 381,159  | 1,916            |
> | **Backward Total**            | **567,245** | **2,678**       |
>
> ---
>
> > **W2: There is a lack of comparison with existing parallel spiking neuron models, such as the PSN family models [1] and PMSN [2] models.**
>
> > **Q3: The existing PSN and PMSN models also aim to address the challenges of parallel training and modeling long-range dependencies. Please discuss the differences between PRF and these models.**
>
> **Reply:** Thank you for pointing out our shortcoming in comparison. In response, we have expanded the comparison with experiments and theoretical analysis. For performance, we have conducted additional experiments on (p)sMNIST, seqCIFAR. As shown in Table.R2, Table.R3 and Table.R4. Within the same architecture, PRF shows highest accuracy compared to other spiking neuron models in psMNIST, seqCifar and listOps.
>
> *Table R2. The (p)seqMNIST (**T=784**) task, all with 4 layer MLP. The results of PMSN and PSN family mainly refer from Cheng X, et al [1]*
> | Source | New Exp. I | New Exp. II | Table.2 | Cheng X, et al [1] | Cheng X, et al | Cheng X, et al | Cheng X, et al | Table.2 |
> |--- |---|---|---|---|---|---|---|---|
> | Neuron          | PRF      | PRF      | PRF     | PMSN [1]   | PSN [2]    | Masked PSN [2]    | Sliding PSN [2]          | LIF     |
> | Architecture            | With residual and Batch Norm (200 channel) | With residual and Batch Norm (128 channel) | w/o residual and BN (128 channel) | With residual and Batch Norm (208 channel) | With residual and Batch Norm (208 channel) | With residual and Batch Norm (208 channel) | With residual and Batch Norm (208 channel) | w/o residual and BN |
> | sMNIST   | 99.39 %  | 99.29 %  | 99.18 % | **99.53 %**    | 97.90 %   | 97.76 %     | 97.20 %    | 89.28 % |
> | psMNIST      | **97.9 %**   | 97.46 %  | 96.67 % | 97.78 %       | 97.76 %   | 97.53 %  | 82.84 %    | 80.36 % |
> | FLOPs   | 246.73M  | 102.9 M  | 102.2 M | 365.8 M        | 712.7 M    | 262.9 M     | 262.9 M     | 244.3 M |
> | Param.     | 167.0k   | 70.4 K   | 69.9 k  | 156.4 k     | 2.5 M      | 153.7 k       | 52.4 k       | 155.1 k |

---

> ### Author Response · Authors · 2024-11-24
> **Reply (2/3)**
>
> *Table R3. The seqCifar (**T=1024**) task. The results of PMSN, PSN and LIF mainly from Cheng X, et al [1]. Since no public code in [1], the specific augmentation techniques used are unclear. On the other hand, [2] employed Mixup and CutMix for seqCIFAR as data augmentation strategies. In contrast, our approach only uses Flip and Cutout as the appropriate data augmentation techniques.*
> | Source     | New Exp. III | New Exp. IV | Cheng X, et al [1] | Cheng X, et al | Cheng X, et al | Cheng X, et al | Cheng X, et al |
> |---|---|---|---|---|---|---|---|
> | Neuron       | PRF         | PRF    | PMSN [1]   | PSN [2]        | Masked PSN [2] | Sliding PSN [2] | LIF    |
> | Arch.    | 6 layer residual block (128 channel) | 6 layer residual block (256 channel) | 6 layer residual block | 6 layer residual block | 6 layer residual block | 6 layer residual block | 6 layer residual block |
> | Acc. (%)   | 82.37%    | **85.33%**     | 82.14%  | 55.24%   | 57.83%   | 70.23%    | 45.07%   |
> |Flops | 584.5 M | 822.65 M | 533.5 M |1640.5 M  | 524.5 M | 524.5 M	| 489.625 M |
> | Param.     | 0.29 M       | 1.1 M      | 0.21 M   | 6.47 M    | 0.38 M   | 0.18 M         | 0.18 M   |
>
> *Table R4. The ListOps (**T=2048**) task. We kept the same training configuration and all hyperparameters, only replacing the neuron. The PSN and Masked PSN [2] do not work for ListOps because these neurons require a fixed T for initialization, whereas the dataset has an unfixed sequence length.*
> | Source       | Table 3 | New Exp. V       |  New Exp. VI          | New Exp. VII |
> |---|----|---|---|---|
> | Neuron  |  PRF  | PSN  [2]   | Masked PSN (k=8) [2]     | Sliding PSN (k=8) [2]  |        |
> | Acc (%)        | **59.20%**       | N.A         | N.A         | 45.90%      |
> | Param. (k)     | 271   | 32.2*1e3           | 32.2*1e3         | 269          |
> | Avg. Train. Time / Epoch |  4 min 10 s        | N.A |  N.A |  24 min 44 s         |
>
> In further, for theoretical analysis, we have also included the overview comparison of PMSN and PSN in Table.R5 and the neuron dynamic and energy consumption comparison in Table.R6. We have include the comparision in Appendix O of manuscript and highlighted in blue for clarity.
>
> The evaluation of the third column of Table R6 is conducted using the common used parameters: sequence length t=1024, the number of hidden units h=128, memory size m=256,  firing rate Fr=0.1, kernel size k=32 (for MPSN, SPSN), and number of compartaments n=4 (for PMSN). For energy consumption analysis, based on the results available in the literature, the energy per accumulation computation E_AC is assumed to be 0.9 pJ, and the energy per multiplication and accumulation E_MAC is assumed to be 4.6 pJ in 45 nm process [3, 4].
>
> *Table R5. The overview features comparision of different neuron models.*
> | Neuron Models     | Parallel Training | Input Cache Free | Oscillation with $ V_{mem} $ | Element-Wise Mul. with Dynamic|
> |---|---|---|---|---|
> | LIF      | No     | **Yes**   | No    | **Yes**     |
> | PSN  [2]      | **Yes**  | No  | No  | -  |
> | Masked PSN [2]   | **Yes** | Partial (When \(k = 1\)) | No  | -  |
> | Sliding PSN [2]    | **Yes**  | Partial (When \(k = 1\)) | No  | -  |
> | PMSN [1]         | Partial  | **Yes** | Partial   | Partial |
> | PRF (Ours)      | **Yes** | **Yes** | **Yes** | **Yes**|
>
>
> *Table R6. The dynamic and thereotical energy. The PRF theoritical energy analysis mainly refer from Appendix E of our manuscript. While other analysis refer from Cheng X et al [1].*
> | **Neuron Models**  | **Dynamics**   | **Theoretical Energy Cost**  | Example Energy / Layer |
> |---|---|---|---|
> | LIF | $V[t] = (1 - \frac{1}{\tau}) V[t-1] + I[t] - \theta S[t-1]$  | $hmtFr_{\mathrm{in}}E_{\mathrm{AC}} + mtE_{\mathrm{MAC}}$                         | 4.22 $ \\mu J$ |
> | PSN    | $V[t] = \sum_{i=0}^t W_{t,i} I[i]$  | $hmtFr_{\mathrm{in}}E_{\mathrm{AC}} + mt^2E_{\mathrm{MAC}}$  | 1237.82 $ \\mu J$ |
> | Masked PSN | $V[t] = \sum_{i=t-k+1}^t W_{t,i} I[i]$  | $hmtFr_{\mathrm{in}}E_{\mathrm{AC}} + kmtE_{\mathrm{MAC}}$  | 41.61 $ \\mu J$ |
> | Sliding PSN | $V[t] = \sum_{i=t-k+1}^t W_{i} I[i]$   | $hmtFr_{\mathrm{in}}E_{\mathrm{AC}} + kmtE_{\mathrm{MAC}}$ | 41.61 $ \\mu J$ |
> | PMSN  | $V_h[t] = \tilde{\tau}V_h[t-1] + \Phi_c I[t]$ | $hmtFr_{\mathrm{in}}E_{\mathrm{AC}} + 8(n-1)mtE_{\mathrm{MAC}}$   | 31.96 $ \\mu J$  |
> | | $I_h[t] = \Phi_b V_h(t) + \gamma_n I(t)$ | | |
> | | $v_s[t] = v_s[t-1] + I_h[t] - \theta S[t-1]$ | | |
> | **PRF (ours)**  | $\tilde{V}[t] = \exp{\left(\Delta \cdot(-\frac{1}{\tau}+i\cdot\theta)\right)} \tilde{V}[t-1] + \Delta I[t]$ | $hmtFr_{\mathrm{in}}E_{\mathrm{AC}} + 5mtE_{\mathrm{Mul}} + 3mtE_{\mathrm{AC}}$ | 8.58 $ \\mu J$  |

---

> ### Author Response · Authors · 2024-11-24
> **Reply (3/3)**
>
> > **Q1: What are the connections between the proposed PRF neuron and SSMs?**
>
> **Reply:** Thank you for your insightfull question. Both PRF and Structured SSMs (State-Space Models) [5] are subsets of SSMs, sharing the same general discretize abstract formulation:
>
>  $ u _ t = \bar{A} u _ {t-1} + \bar{B} c _ t, \quad y _ t = f(u _ t). $
>
> The main differences is the computation of A, B and f. The comparison with SSM is detailed in Table R7. And we added the new table to the Appendix O.4 of revised manuscript.
>
> *Table R7. Comparison Between PRF and Structured SSMs.*
> | **Component** | **Structured SSMs**    | **PRF**    |
> |---|---|---|
> | $\bar{A}$     | $\exp(\Delta A) \in \mathbb{R}^{h \times h}$    | $\exp(\Delta \cdot (-\frac{1}{\tau} + i \theta)) \in \mathbb{C}^d$   |
> | $\bar{B}$     | $A^{-1} \left(e^{A \Delta} - I \right) B \in \mathbb{R}^{h \times d}$     | $\Delta \in \mathbb{R}^d$         |
> | $f(u_t)$      | $C \cdot u_t$    | $\mathcal{H}(\Re(u_t) - V_{th})$      |
> |      | $C \in \mathbb{R}^{d \times h}, f: \mathbb{R}^{h} \to \mathbb{R}^{d}$ | $f: \mathbb{R}^d \to \mathbb{R}^d$ |
>
> > **Q2. What impact does the reset have on the performance of the PRF model in LRA tasks?**
>
> **Reply:** We sincerely appreciate your insightful question. We should have clarified the impact of the reset mechanism on performance more clearly. In response to your feedback, we conducted an additional experiment (ListOps of LRA tasks) by reintroducing a reset mechanism into the PRF neuron. The results are summarized in Table R8. PRF neurons are designed to oscillate the membrane potential, thereby eliminating the need for an explicit reset process. This oscillation is regarded as an implicit reset mechanism. The introduction of an additional explicit reset limits the neurons' capability to oscillate, which causes performance degradation. At the same time, the training time also increases with the introduction of the additional explicit reset.
>
> *Table R8. The impact of reset on the performance of PRF.*
> |ListOps (T=2048) | Task Acc.	| Training time / epoch |
> | --- | --- | --- |
> |PRF | 59.2 % | 4 min 10 s |
> | PRF + reset | 37.2 % | 40 min 28 s |
>
> ---
>
> Reference:
>
> [1] Chen X, Wu J, Ma C, et al. PMSN: A Parallel Multi-compartment Spiking Neuron for Multi-scale Temporal Processing. arXiv preprint arXiv:2408.14917, 2024.
>
> [2] Fang W, Yu Z, Zhou Z, et al. Parallel spiking neurons with high efficiency and ability to learn long-term dependencies. NeurIPS, 2024, 36.
>
> [3] Han S, Pool J, Tran J, et al. Learning both weights and connections for efficient neural network. NeurIPS, 2015, 28.
>
> [4] Mark Horowitz. Energy table for 45nm process, Stanford VLSI wiki.
>
> [5] Gu A, Goel K, Re C. Efficiently Modeling Long Sequences with Structured State Spaces. ICLR, 2022.
>
> ---
>
> We sincerely appreciate and thank you again for your valuable review and feedback. If any concerns or questions arise during the discussion period, we would be happy to provide further clarification. We would also be grateful if you will consider a more positive evaluation of our work once all concerns have been addressed.
>
>
> -- Authors

---

> > ### Comment · Reviewer_ZwjR · 2024-11-26
> >
> > I appreciate the authors' detailed response, which has addressed my concerns regarding comparisons with existing parallel spiking neuron models and the relationship with SSMs. Considering the comprehensive experiments, thorough analysis, and performance improvements, particularly on the LRA tasks, I have raised my score to 8.

---

### Official Review · Reviewer_B3wb · 2024-11-04

**Soundness:** 3
**Presentation:** 3
**Contribution:** 3
**Rating:** 6
**Confidence:** 4

**Summary:**

The authors introduce a new spiking neuron model, the parallel resonate and fire (PRF) neuron, that has two advantages over the classic leaky integrate-and-fire (LIF) neuron.

1) it is parallelizable and thus speeds up training and inference times

2) it has a longer-term memory because it can resonate.

**Strengths:**

The accuracy on the LRA benchmark is trully impressive, almost as good as S4, while it could consume much less energy (at least in theory).

**Weaknesses:**

* The accuracy on spMNIST is much less convicing. Here two similar recent proposals are much more accurate: Li et al 2024 (cited) and Bal et al 2024 (cited)
These refs should be added to Table 2.
I also suggest adding sequential CIFAR, which is more challenging.

* Other parallel spiking neuron models should be cited and added to Fig 4 left and to the comparison tables when possible, and the differences/similarities should be highlighted:  Fang et al 2023 (cited), Chen et al 2024, Yarga et al 2024

* The resonance seems similar to Baronig et al. 2024, which should be cited, and the differences/similarities should be highlighted.

References:

Baronig M, Ferrand R, Sabathiel S, Legenstein R (2024) Advancing Spatio-Temporal Processing in Spiking Neural Networks through Adaptation. arXiv Available at: http://arxiv.org/abs/2408.07517.

Fang W, Yu Z, Zhou Z, Chen D, Chen Y, Ma Z, Masquelier T, Tian Y (2023) Parallel Spiking Neurons with High Efficiency and Ability to Learn Long-term Dependencies. NeurIPS 1:1–13 Available at: http://arxiv.org/abs/2304.12760.

Zeyu Liu, Gourav Datta, Anni Li, and Peter Anthony Beerel. Lmuformer: Low complexity yet powerful spiking model with legendre memory units. arXiv preprint arXiv:2402.04882, 2024.

Bal M, Sengupta A (2024) P-SpikeSSM: Harnessing Probabilistic Spiking State Space Models for Long-Range Dependency Tasks. arXiv:1–16 Available at: http://arxiv.org/abs/2406.02923.

Chen X, Wu J, Ma C, Yan Y, Wu Y, Tan KC (2024) PMSN: A Parallel Multi-compartment Spiking Neuron for Multi-scale Temporal Processing. arXiv Available at: http://arxiv.org/abs/2408.14917.

Yarga SYA, Wood SUN (2024) Accelerating Spiking Neural Networks with Parallelizable Leaky Integrate-and-Fire Neurons. TechRxiv:1–10.

--

POST REBUTTAL:

All my main concerns have been addressed, so I'm raising my score to 6. I don't go higher because similar parallel neurons have been proposed recently and because I still think the membrane potential (float) residual connections could be an issue for some neuromorphic chips. Still, I think this paper should be accepted.

**Questions:**

* Any idea why Li et al 2024 does better than you on psMNIST, but not on LRA?

* L244 "At this point, all recursive forms have been converted into dynamic equations." I agree, but still, the dynamic equations should presumably be solved sequentially. So, I don't see how this helps parallelization. More insight is needed here.

* According to Fig 2 the residual connections carry floats, not spikes. This could be a pb for implementation on neuromorphic chips. Could the authors elaborate?

* L505 "This energy reduction mainly benefits from the extreme sparsity of spikes"
Could even higher levels of sparsity could be encouraged via some additional terms in the loss function?

---

> ### Author Response · Authors · 2024-11-24
> **Reply (1/3)**
>
> Dear reviewer B3wb,
>
> We sincerely appreciate your valuable feedback and thoughtful suggestions. We greatly appreciate the time and effort you have dedicated to evaluating our work. To address your concerns, we will respond to each of the weaknesses and questions you raised individually and in detail.
>
> ----
>
> > **W1: The accuracy on spMNIST is much less convicing. Here two similar recent proposals are much more accurate: Li et al 2024 (cited) and Bal et al 2024 (cited) These refs should be added to Table 2. I also suggest adding sequential CIFAR, which is more challenging.**
>
> > **Q1. Any idea why Li et al 2024 does better than you on psMNIST, but not on LRA?”**
>
> **Reply:** Thank you for your feedback and for highlighting these references. To address your concerns, we conducted additional experiments on both psMNIST and sequential CIFAR for a more comprehensive evaluation. The results are shown in Tables R1 and R2 and are summarized below:
>
> - On psMNIST, we achieved performance comparable to Li et al. (2024) [1] and Bal et al. (2024) [2], while significantly reducing computational FLOPs.
>
>     1. Comparison with Li et al. (2024): By increasing the size of our 4-layer MLP with PRF neurons (New Exp I), we achieved accuracy and parameter sizes comparable to Li et al. (2024) [1], while reducing computational FLOPs by half. This highlights the computational efficiency of our PRF neurons.
>     2. Comparison with Bal et al. (2024): We constructed a 2-layer SD-TCM with PRF neurons (New Exp II), achieving accuracy and parameter sizes similar to Bal et al. (2024) [2], while reducing computational FLOPs by over 80%.
>
> -  On sequential CIFAR, our model outperformed PSN family models while maintaining lower computational costs, further demonstrating the efficiency and versatility of our PRF neurons.
>
> We will add these results and comparisons at the appropriate location in the final revised manuscript.
>
> *Table R1. The comparision of ps-MINIST (T=784) task*
> | Source  | SpikingLMUFormer (Li et al 2024) [1] | p-SpikeSSM (Bal et al 2024) [2] | Table 2 in manuscript | New Exp. I | New Exp. II |
> |---|---|---|---|---|---|
> | **Arch** | 1-layer LMUFormer | 2-layer SSM (400 Channel)   | 4L-MLP (128 Channel)   | 4L-MLP (200 Channel)  | 2-layer SD-TCM (400 Channel) |
> | **Neuron**   | LIF | Stoch. Neuron  | PRF      | PRF     | PRF  |
> | **psMNIST**       | 97.92 %         | 98.40 %         | 96.83 %               | 97.9 %                | 98.11 %                  |
> | **FLOPs**       | 576.3 M       | 5,053.7 M     | 102.2 M    | 246.7 M       | 963.6 M                  |
> | **Parameters**    | 166.1 k (extract from public code)  | No Public source     | 68.9 k     | 167.0 k        | 640.0 k      |
>
>
> *Table R2. The seqCifar (**T=1024**) task. The results of PMSN, PSN and LIF mainly from Cheng X, et al [1]. Since no public code in [1], the specific augmentation techniques used are unclear. On the other hand, [2] employed Mixup and CutMix for seqCIFAR as data augmentation strategies. In contrast, our approach only uses Flip and Cutout as the appropriate data augmentation techniques.*
> | Source     | New Exp. III | New Exp. IV | Cheng X, et al [3] | Cheng X, et al | Cheng X, et al | Cheng X, et al | Cheng X, et al |
> |---|---|---|---|---|---|---|---|
> | Neuron       | PRF         | PRF    | PMSN [3]   | PSN [4]        | Masked PSN [4] | Sliding PSN [4] | LIF    |
> | Arch.    | 6 layer residual block (128 channel) | 6 layer residual block (256 channel) | 6 layer residual block | 6 layer residual block | 6 layer residual block | 6 layer residual block | 6 layer residual block |
> | Acc. (%)   | 82.37%    | **85.33%**     | 82.14%  | 55.24%   | 57.83%   | 70.23%    | 45.07%   |
> |Flops | 584.5 M | 822.65 M | 533.5 M |1640.5 M  | 524.5 M | 524.5 M	| 489.625 M |
> | Param.     | 0.29 M       | 1.1 M      | 0.21 M   | 6.47 M    | 0.38 M   | 0.18 M         | 0.18 M   |

---

> ### Author Response · Authors · 2024-11-24
> **Reply (2/3)**
>
> ----
>
> > **W2. Other parallel spiking neuron models should be cited and added to Fig 4 left and the comparison tables when possible, and the differences/similarities should be highlighted: to Fang et al 2023 (cited), Chen et al 2024, Yarga et al 2024**
>
> > **W3. The resonance seems similar to Baronig et al. 2024, which should be cited, and the differences/similarities should be highlighted.**
>
> **Reply:** Thank you for providing the references and helpful suggestions. We will add the comparison with PSN in Figure 4 and Table 2 in the revised manuscript. We would also like to emphasize that the primary objective of the psMNIST experiment is to evaluate and compare the performance of different neuron models within a simple architecture (4-layer MLP with 128 channels), rather than solely focusing on achieving the highest possible accuracy. As shown in Table R1, we achieved an accuracy of 98.11\% with the 2-layer SD-TCM on psMNIST, and we will incorporate these new results into the revised Table 2, along with the references.
>
> We compare the distinct features of various neuron models, as shown in Table R3. Additionally, we analyze their dynamics alongside the corresponding theoretical energy costs in Table R4. These tables have been added to Appendix O of the manuscript to clarify this comparison.
>
> The evaluation in the third column of Table R4 is conducted using commonly used parameters: sequence length $t = 1024$, number of hidden units $h = 128$, memory size $m = 256$, firing rate $Fr = 0.1$, kernel size $k = 32$ (for MPSN, SPSN), and number of compartments $n = 4$ (for PMSN). For the energy consumption analysis, based on results available in the literature, the energy per accumulation computation $E _ {\text{AC}}$ is assumed to be 0.9 pJ, and the energy per multiplication and accumulation $E _ {\text{MAC}}$ is assumed to be 4.6 pJ in a 45 nm process [7, 8]. This analysis shows that the PRF neurons have the lowest energy consumption compared to other parallel neurons.
>
> *Table R3. The overview features comparision of different neuron models.*
> | Neuron Models     | Parallel Training | Input Cache Free | Oscillation with $ V _ {mem} $ | Element-Wise Mul. with Dynamic|
> |---|---|---|---|---|
> | LIF      | No     | **Yes**   | No    | **Yes**     |
> | PSN  [4]      | **Yes**  | No  | No  | -  |
> | Masked PSN [4]   | **Yes** | Partial (When \(k = 1\)) | No  | -  |
> | Sliding PSN [4]    | **Yes**  | Partial (When \(k = 1\)) | No  | -  |
> | PMSN [3]         | Partial  | **Yes** | Partial   | Partial |
> | adLIF  [5]  | No | **Yes** | **Yes**| **Yes** |
> | Parallelizable LIF [6] | **Yes** | **Yes** | No | **Yes** |
> | PRF (Ours)      | **Yes** | **Yes** | **Yes** | **Yes**|
>
>
> *Table R4. The dynamic and thereotical energy. The PRF theoritical energy analysis mainly refer from Appendix E of our manuscript. While other analysis refer from Cheng X et al [1].*
> | **Neuron Models**  | **Dynamics**   | **Theoretical Energy Cost**  | Example Energy / Layer |
> |---|---|---|---|
> | LIF | $ V[t] = (1 - \frac{1}{\tau}) V[t-1] + I[t] - \theta S[t-1] $  | $ hmtFr _ {\mathrm{in}} E _ {\mathrm{AC}} + m t E _ {\mathrm{MAC}}$                         | 4.22 $ \\mu J$ |
> | PSN [4]   | $V[t] = \sum _ {i=0} ^ t W _ {t,i} I[i]$  | $ h m t Fr _ {\mathrm{in}} E _ {\mathrm{AC}} + mt ^ 2 E _ {\mathrm{MAC}}$  | 1237.82 $ \\mu J$ |
> | Masked PSN [4]| $ V[t] = \sum _ {i = t - k + 1} ^ t W _ {t,i} I[i]$  | $hmtFr _ {\mathrm{in}}E _ {\mathrm{AC}} + kmtE _ {\mathrm{MAC}}$  | 41.61 $ \\mu J$ |
> | Sliding PSN [4]| $ V[t] = \sum _ {i=t-k+1}^t W _ {i} I[i]$   | $hmtFr _ {\mathrm{in}}E _ {\mathrm{AC}} + kmtE _ {\mathrm{MAC}}$ | 41.61 $ \\mu J$ |
> | PMSN  [3] | $V _ h[t] = \tilde{\tau}V _ h[t-1] + \Phi _ c I[t]$ | $hmtFr _ {\mathrm{in}} E _ {\mathrm{AC}} + 8(n-1) mtE _ {\mathrm{MAC}}$   | 31.96 $ \\mu J$  |
> | | $ I _ h[t] = \Phi _ b V _ h(t) + \gamma _ n I(t)$ | | |
> | | $ v _ s[t] = v _ s[t-1] + I _ h[t] - \theta S[t-1]$ | | |
> | adLIF [5] | $\hat{u}[t] = \alpha u[t-1] + (1-\alpha)(-w[t-1] + I[t])$ | $hmtFr _ {\mathrm{in}}E _ {\mathrm{AC}} + 6mtE _ {\mathrm{MAC}}$  | 10.26 $ \\mu J$  |
> | | $w[t] = \beta w[t-1] + (1-\beta)\left(a\hat{u}[t-1]\cdot(1 - S[t-1]) + bS[t]\right)$ | | |
> | **PRF (ours)**  | $\tilde{V}[t] = \exp{\left(\Delta \cdot(-\frac{1}{\tau}+i\cdot\theta)\right)} \tilde{V}[t-1] + \Delta I[t]$ | $h m t Fr _ {\mathrm{in}} E _ {\mathrm{AC}} + 5 m t E _ {\mathrm{Mul}} + 3 m t E _ {\mathrm{AC}}$ | 8.58 $ \\mu J$  |

---

> > ### Author Response · Authors · 2024-11-24
> > **Reply (3/3)**
> >
> > > **Q2. L244 "At this point, all recursive forms have been converted into dynamic equations." I agree, but still, the dynamic equations should presumably be solved sequentially. So, I don't see how this helps parallelization. More insight is needed here.**
> >
> > **Reply:** Thank you again for your valuable question. We agree with the reviewer that the dynamic equations are solved sequentially. Nevertheless, our approach enables the input and output operations to proceed in parallel. This eliminates the need for the membrane potential update to wait for the previous spike outputs, thereby enhancing computational efficiency during both the forward and backward passes. We have listed the time for each block with LIF and Parallelized LIF in the seqMNIST task, as shown in Table R5.
> >
> > The forward pass benefits from a significant reduction in charge and fire time, with parallelization reducing the total forward runtime by approximately 43\%. This improvement occurs because the dynamic equations eliminate the need to wait for the previous timestep's spike output during the membrane potential update. Note that the reset of dynamic equation could theoretically be optimized from $O(L)$ to $O(L \log L)$, since only element-wise multiplication and comparison operations are involved during the dynamic equation computation.
> >
> > Furthermore, the backward pass shows significant speedups, with the total backward time reduced from 567,245 $\mu$s to just 2,678 $\mu$s. These improvements are due to parallelizing the LIF input and output, which reduces the complexity of recursive graph construction during the backward pass (auto-grad) compared to sequential LIF. We hope this clarifies the benefits of our parallelization approach.
> >
> > *Table R5. The runtime of LIF and Parallelized LIF with 784 sequence length inputs.*
> > | Run Time ($\mu s$) | LIF      | Parallelized LIF |
> > |---|---|---|
> > | Forward: Charge | 146,725  | 60,261|
> > | Forward: Reset | 150,120  | 158,774 |
> > | Forward: Fire | 91,398   | 459 |
> > | **Forward Total** | **388,240** | **219,494** |
> > | Backward: Fire (Surrogate) | 186,086  | 762 |
> > | Backward: Membrane Part | 381,159  | 1,916 |
> > | **Backward Total** | **567,245**  | **2,678** |
> >
> > ----
> >
> > > **Q3. According to Fig 2 the residual connections carry floats, not spikes. This could be a pb for implementation on neuromorphic chips. Could the authors elaborate?**
> >
> > **Reply:** Thank you for your insightful question. The residual connection we used is a membrane shortcut residual [9], which could be suitable for implementation on neuromorphic chips, as illustrated in previous work [10]. This type of residual does not affect the addition operation in the subsequent Spike-Linear computations, and the membrane shortcut residual only requires the addition operation.
> >
> > ----
> >
> > > **Q4. L505 "This energy reduction mainly benefits from the extreme sparsity of spikes" Could even higher levels of sparsity could be encouraged via some additional terms in the loss function?**
> >
> > **Reply:** Thank you for your insightful question. In our work, the average firing rate of PRF is lower than 5% in LRA tasks, as shown in Table 11 in Appendix M of the manuscript. This contrasts with the firing rate of LIF neuron around 10% in the literature [11, 12]. We tentatively attribute our lower firing rate to the fact that the oscillating membrane potential more frequently reaches negative values compared to the commonly used LIF model. We have not incorporated any additional loss function to further reduce the sparsity. Theoretically, higher sparsity could be encouraged via an additional loss function, but we plan to investigate this aspect in our future studies.

---

> > > ### Author Response · Authors · 2024-11-24
> > >
> > > ---
> > >
> > > Reference:
> > >
> > > [1] Liu, et al. LMUformer: Low complexity yet powerful spiking model with Legendre memory units. ICLR, 2024.
> > >
> > > [2] Bal, et al. P-SpikeSSM: Harnessing Probabilistic Spiking State Space Models for Long-Range Dependency Tasks. arXiv: http://arxiv.org/abs/2406.02923.
> > >
> > > [3] Chen, et al.PMSN: A Parallel Multi-compartment Spiking Neuron for Multi-scale Temporal Processing. arXiv Avail
> > >
> > > [4] Fang, et al. Parallel Spiking Neurons with High Efficiency and Ability to Learn Long-term Dependencies. NeurIPS, 2024: http://arxiv.org/abs/2304.12760.
> > >
> > > [5] Baronig M, Ferrand R, Sabathiel S, Legenstein R (2024) Advancing Spatio-Temporal Processing in Spiking Neural Networks through Adaptation. arXiv Available at: http://arxiv.org/abs/2408.07517.
> > >
> > > [6] Yarga SYA, Wood SUN (2024) Accelerating Spiking Neural Networks with Parallelizable Leaky Integrate-and-Fire Neurons. TechRxiv:1–10.
> > >
> > > [7] Han S, Pool J, Tran J, et al. Learning both weights and connections for efficient neural network. NeurIPS, 2015, 28.
> > >
> > > [8] Mark Horowitz. Energy table for 45nm process, Stanford VLSI wiki.
> > >
> > > [9] Hu et al. Advancing spiking neural networks toward deep residual learning. TNNLS, 2024
> > >
> > > [10] Yao et al. Spike-Driven Transformer V2: Meta Spiking Neural Network Architecture Inspiring the Design of Next-generation Neuromorphic Chips. ICLR 2024.
> > >
> > > [11] Yao et al. Spike-Driven Transformer. NeurIPS, 2024.
> > >
> > > [12] Fang et al. Deep Residual Learning in Spiking Neural Networks. NeurIPS, 2021.
> > >
> > > ---
> > >
> > > We sincerely appreciate and thank you again for your valuable review and feedback. If any concerns or questions arise during the discussion period, we would be happy to provide further clarification. We would also be grateful if you could consider a more positive evaluation of our work once all concerns have been addressed.
> > >
> > > -- Authors

---

### Author Response · Authors · 2024-12-03

We extend our sincere appreciation to the four reviewers. Your valuable reviews have greatly improved the quality of our manuscript. We also appreciate the constructive suggestions during the rebuttal and discussion phases.

To summarize, our contribution is the introduction of a decoupled reset mechanism to accelerate training and inference for long-sequence learning. We designed a PRF neuron by incorporating the reset part as an imaginary component, which enhances long-range dependency capabilities while maintaining computational efficiency. Finally, the SD-TCM architecture, tailored for PRF neurons, achieves performance comparable to S4 on the LRA dataset while saving 98.57% of energy.

During the rebuttal phase, we extended experiments to compare PRF neurons with other neuron types, showing that PRF outperforms them on the psMNIST, sCifar, and LRA tasks. Although PRF performs strongly in the specially designed SD-TCM architecture for long-sequence tasks, it can also be applied to simple MLP architectures as well as more complex SSM architectures. We would like to express our gratitude to the reviewers again for their valuable feedback on our manuscript.

---

### Meta-Review · Area_Chair_GiuG · 2024-12-26

**Metareview:**

This paper proposes a parallel resonate and fire (PRF) neuron model aimed at efficient long-sequence learning. The main claims are: 1) A decoupled reset method that reduces training time from O(L²) to O(LlogL), 2) A PRF neuron design that leverages oscillating membrane potential for capturing long-range dependencies, and 3) A spike-driven architecture using PRF that achieves performance comparable to S4 while reducing energy consumption by two orders of magnitude.

Positive:

- Novel approach to parallel training in SNNs through the decoupled reset

Weaknesses:

- reproducibility concerns due to lack of code availability at submission time
- limited exploration of PRF neuron effectiveness across different architectures
- overlooked the most relevant state-of-the-art methods to compare against: Liquid-S4 (https://arxiv.org/abs/2209.12951) and the entire line of Liquid time-constant networks (https://arxiv.org/abs/2006.04439) work, that shares high degrees of similarities in scope with this model.

I vote for the rejection of the paper.

**Additional Comments On Reviewer Discussion:**

The rebuttal period focused on three main concerns: (1) lack of comprehensive ablation studies, (2) limited evaluation across different architectures, and (3) reproducibility concerns. The authors responded with additional experiments, including ablation studies on SD-TCM and evaluations on MLP and SSM architectures. While these additions were helpful, they actually revealed significant limitations of the method, particularly the overfitting issues with SSMs and the heavy dependence on the specific architecture. The authors also provided simplified PyTorch-like implementation, but this came late in the process and does not fully address reproducibility concerns.

---

### Decision · Program_Chairs · 2025-01-22

Reject